

# Microbial methanogenesis in the sulfate-reducing zone in sediments
# from Eckernförde Bay, SW Baltic Sea
Johanna Maltby[1,2*], Lea Steinle[3,1], Carolin R. Löscher[4,1], Hermann W. Bange[1], Martin A. Fischer[5], Mark
Schmidt[1], Tina Treude[1,6*]
[1] GEOMAR Helmholtz Centre for Ocean Research Kiel, Department of Marine Biogeochemistry, 24148
Kiel, Germany
[2] Present Address: Natural Sciences Department, Saint Joseph's College, Standish, Maine 04084, USA
[3] Department of Environmental Sciences, University of Basel, 4056 Basel, Switzerland
[4] Nordic Center for Earth Evolution, University of Southern Denmark, 5230 Odense, Denmark
[5] Institute of Microbiology, Christian-Albrecht-University Kiel, 24118 Kiel, Germany
[6] Department of Earth, Planetary, and Space Sciences, Department of Atmospheric and Oceanic
Sciences, University of California, Los Angeles (UCLA), Los Angeles, California 90095-1567, USA
*Correspondence: jmaltby@sjcme.edu, ttreude@g.ucla.edu





## Abstract

The presence of surface methanogenesis, located within the sulfate-reducing zone (0-30 centimeters below seafloor, cmbsf), was investigated in sediments of the seasonally hypoxic Eckernförde Bay, southwestern Baltic Sea. Water column parameters like oxygen, temperature and salinity together with porewater geochemistry and benthic methanogenesis rates were determined in the sampling area "Boknis Eck" quarterly from March 2013 to September 2014, to investigate the effect of seasonal environmental changes on the rate and distribution of surface methanogenesis and to estimate its potential contribution to benthic methane emissions. The metabolic pathway of methanogenesis in the presence or absence of sulfate reducers and after the addition of a non-competitive substrate was studied in four experimental setups: 1) unaltered sediment batch incubations (net methanogenesis), 2) $^{14}$C-bicarbonate labeling experiments (hydrogenotrophic methanogenesis), 3) manipulated experiments with addition of either molybdate (sulfate reducer inhibitor), 2-bromoethane-sulfonate (methanogen inhibitor), or methanol (non-competitive substrate, potential methanogenesis), 4) addition of $^{13}$C-labeled methanol (potential methylotrophic methanogenesis). After incubation with methanol in the manipulated experiments, molecular analyses were conducted to identify key functional methanogenic groups. Hydrogenotrophic methanogenesis in sediments below the sulfate-reducing zone (> 30 cmbsf) was determined by $^{14}$C-bicarbonate radiotracer incubation in samples collected in September 2013.

Surface methanogenesis changed seasonally in the upper 30 cmbsf with rates increasing from March (0.2 nmol cm$^{-3}$ d$^{-1}$) to November (1.3 nmol cm$^{-3}$ d$^{-1}$) 2013 and March (0.2 nmol cm$^{-3}$ d$^{-1}$) to September (0.4 nmol cm$^{-3}$ d$^{-1}$) 2014, respectively. Its magnitude and distribution appeared to be controlled by organic matter availability, C/N, temperature, and oxygen in the water column, revealing higher rates in warm, stratified, hypoxic seasons (September/November) compared to colder, oxygenated seasons (March/June) of each year. The majority of surface methanogenesis was likely driven by the usage of non-competitive substrates (e.g., methanol and methylated compounds), to avoid competition with sulfate reducers, as it was indicated by the 1000-3000-fold increase in potential methanogenesis activity observed after methanol addition. Accordingly, competitive hydrogenotrophic methanogenesis increased in the sediment only below the depth of sulfate penetration (> 30 cmbsf). Members of the family *Methanosarcinaceae,* which are known for methylotrophic methanogenesis, were detected by PCR using *Methanosarcinaceae*-specific primers and are likely to be responsible for the observed surface methanogenesis.

The present study indicated that surface methanogenesis makes an important contribute to the benthic methane budget of Eckernförde Bay sediments as it could directly feed into methane oxidation above the sulfate-methane transition zone.





## 1. Introduction

After water vapor and carbon dioxide, methane is the most abundant greenhouse gas in the
atmosphere (e.g. Hartmann et al., 2013; Denman et al., 2007). Its atmospheric concentration
increased more than 150 % since preindustrial times, mainly through increased human activities such
as fossil fuel usage and livestock breeding (Hartmann et al., 2013; Wuebbles & Hayhoe, 2002;
Denman et al., 2007). Determining the natural and anthropogenic sources of methane is one of the
major goals for oceanic, terrestrial and atmospheric scientists to be able to predict further impacts
on the world's climate. The ocean is considered to be a modest natural source for atmospheric
methane (Wuebbles & Hayhoe, 2002; Reeburgh, 2007; EPA, 2010). However, research is still sparse
on the origin of the observed oceanic methane, which automatically leads to uncertainties in current
ocean flux estimations (Bange et al., 1994; Naqvi et al., 2010; Bakker et al., 2014).

Within the marine environment, the coastal areas (including estuaries and shelf regions) are
considered the major source for atmospheric methane, contributing up to 75 % to the global ocean
methane production (Bange et al., 1994). The major part of the coastal methane is produced during
microbial methanogenesis in the sediment, with probably only a minor part originating from
methane production within the water column (Bakker et al., 2014). However, the knowledge on
magnitude, seasonality and environmental controls of benthic methanogenesis is still limited.

In marine sediments, methanogenesis activity is mostly restricted to the sediment layers below
sulfate reduction, due to the successful competition of sulfate reducers with methanogens for the
mutual substrates acetate and hydrogen ($H_2$) (Oremland & Polcin, 1982; Crill & Martens, 1986;
Jørgensen, 2006). Methanogens produce methane mainly from using acetate (acetoclastic
methanogenesis) or $H_2$ and carbon dioxide ($CO_2$) (hydrogenotrophic methanogenesis). Competition
with sulfate reducers can be relieved through usage of non-competitive substrates (e.g. methanol or
methylated compounds, methylotrophic methanogenesis) (Cicerone & Oremland, 1988; Oremland &
Polcin, 1982). Coexistence of sulfate reduction and methanogenesis has been detected in a few
studies from organic-rich sediments, e.g., salt-marsh sediments (Oremland et al., 1982; Buckley et al.,
2008), coastal sediments (Holmer & Kristensen, 1994; Jørgensen & Parkes, 2010) or sediments in
upwelling regions (Pimenov et al., 1993; Ferdelman et al., 1997; Maltby et al., 2016), indicating the
importance of these environments for surface methanogenesis. So far, however, environmental
control mechanisms of surface methanogenesis remain elusive.

The coastal inlet Eckernförde Bay (southwestern Baltic Sea) is an excellent model environment to
study seasonal and environmental control mechanisms of benthic surface methanogenesis. Here,
the muddy sediments are characterized by high organic loading and high sedimentation rates
(Whiticar, 2002), which lead to anoxic conditions within the uppermost 0.1-0.2 centimeter below
seafloor (cmbsf) (Preisler et al., 2007). Seasonally hypoxic (dissolved oxygen < 63 μM) and anoxic



(dissolved oxygen = 0 µM) events in the bottom water of Eckernförde Bay (Lennartz et al., 2014)
provide ideal conditions for anaerobic processes at the sediment surface.
Sulfate reduction is the dominant pathway of organic carbon degradation in Eckernförde Bay
sediments in the upper 30 cmbsf, followed by methanogenesis in deeper sediment layers where
sulfate is depleted (> 30 cmbsf) (Whiticar 2002; Treude et al. 2005; Martens et al. 1998). This deep
methanogenesis can be intense and often leads to methane oversaturation in the porewater below
50 cm sediment depth, resulting in gas bubble formation (Abegg & Anderson, 1997; Whiticar, 2002;
Thießen et al., 2006). Thus, methane is transported from the methanogenic zone (> 30 cmbsf) to the
surface sediment by both molecular diffusion and advection via rising gas bubbles (Wever et al.,
1998; Treude et al., 2005a). Although upward diffusing methane is mostly retained by anaerobic
oxidation of methane (AOM) (Treude et al. 2005), a major part is reaching the sediment-water
interface through gas bubble transport (Treude et al. 2005; Jackson et al. 1998), resulting in a
supersaturation of the water column with respect to atmospheric methane concentrations (Bange et
al., 2010). The Time Series Station "Boknis Eck" in the Eckernförde Bay is a known site of methane
emissions into the atmosphere throughout the year due to this supersaturation of the water column
(Bange et al., 2010).
The source for benthic and water column methane was seen in deep methanogenesis (> 30 cmbsf)
below the penetration of sulfate (Whiticar, 2002), however, coexistence of sulfate reduction and
methanogenesis has been postulated (Whiticar, 2002; Treude et al., 2005a). Still, the magnitude and
environmental controls of surface methanogenesis is poorly understood, even though it may make a
measurable contribution to benthic methane emissions given its short diffusion distance to the
sediment-water interface (Knittel & Boetius, 2009). Production of methane within the sulfate
reduction zone of Eckernförde Bay surface sediments could further explain peaks of methane
oxidation observed in top sediment layers, which was previously attributed to methane transported
to the surface via rising gas bubbles (Treude et al., 2005a).
In the present study, we investigated surface sediment (< 30 cmbsf, on a seasonal basis), deep
sediment (> 30 cmbsf, on one occasion), and the water column (on a seasonal basis) at the Time
Series Station "Boknis Eck" in Eckernförde Bay, to validate the existence of surface methanogenesis
and its potential contribution to benthic methane emissions. Water column parameters like oxygen,
temperature, and salinity together with porewater geochemistry and benthic methanogenesis were
measured over a course of 2 years. In addition to seasonal rate measurements, inhibition and
stimulation experiments, stable isotope probing, and molecular analysis were carried out to find out
if surface methanogenesis 1) is controlled by environmental parameters, 2) shows seasonal
variability, 3) is based on non-competitive substrates with a special focus on methylotrophic
methanogens.



## 2. Material and Methods

### 2.1 Study site

Samples were taken at the Time Series Station "Boknis Eck" (BE, 54°31.15 N, 10°02.18 E; www.bokniseck.de) located at the entrance of Eckernförde Bay in the southwestern Baltic Sea with a water depth of about 28 m (map of sampling site can be found in e.g. Hansen et al., (1999)). From mid of March until mid of September the water column is strongly stratified due to the inflow of saltier North Sea water and a warmer and fresher surface water (Bange et al., 2011). Organic matter degradation in the deep layers causes pronounced hypoxia (March-Sept) or even anoxia (August/September) (Smetacek, 1985; Smetacek et al., 1984). The source of organic material is phytoplankton blooms, which occur regularly in spring (February-March) and fall (September-November) and are followed by pronounced sedimentation of organic matter (Bange et al., 2011). To a lesser extent, phytoplankton blooms and sedimentation are also observed during the summer months (July/August) (Smetacek et al., 1984). Sediments at BE are generally classified as soft, fine-grained muds (< 40 μm) with a carbon content of 3 to 5 wt% (Balzer et al., 1986). The bulk of organic matter in Eckernförde Bay sediments originates from marine plankton and macroalgal sources (Orsi et al., 1996), and its degradation leads to production of free methane gas (Wever & Fiedler, 1995; Abegg & Anderson, 1997; Wever et al., 1998). The oxygen penetration depth is limited to the upper few millimeters when bottom waters are oxic (Preisler et al., 2007). Reducing conditions within the sulfate reduction zone lead to a dark grey/black sediments color with a strong hydrogen sulfur odor in the upper meter of the sediment and dark olive-green color the deeper sediment layers (> 1 m) (Abegg & Anderson, 1997).

### 2.2 Water column and sediment sampling

Sampling was done on a seasonal basis during the years of 2013 and 2014. One-Day field trips with either F.S. Alkor (cruise no. AL410), F.K. Littorina or F.B. Polarfuchs were conducted in March, June, and September of each year. In 2013, additional sampling was conducted in November. At each sampling month, water profiles of temperature, salinity, and oxygen concentration (optical sensor, RINKO III, detection limit= 2 μM) were measured with a CTD (Hydro-Bios). In addition, water samples for methane concentration measurements were taken at 25 m water depth with a 6-Niskin bottle (4 Liter each) rosette attached to the CTD (Table 1). Complementary samples for water column chlorophyll were taken at 25 m water depth with the CTD-rosette within the same months during standardized monthly sampling cruises to Boknis Eck organized by GEOMAR.

Sediment cores were taken with a miniature multicorer (MUC, K.U.M. Kiel), holding 4 core liners (length= 60 cm, diameter= 10 cm) at once. The cores had an average length of ~ 30 cm and were





stored at 10°C in a cold room (GEOMAR) until further processing (normally within 1-3 days after
sampling).
In September 2013, a gravity core was taken in addition to the MUC cores. The gravity core was
equipped with an inner plastic bag (polyethylene; diameter: 13 cm). After core recovery (330 cm
total length), the polyethylene bag was cut open at 12 different sampling depths resulting in intervals
of 30 cm and sampled directly on board for sediment porewater geochemistry (see Sect. 2.4),
sediment methane (see Sect. 2.5), sediment solid phase geochemistry (see Sect. 2.6), and microbial
rate measurements for hydrogenotrophic methanogenesis as described in section 2.8.
**2.3 Water column parameters**
At each sampling month, water samples for methane concentration measurements were taken at 25
m water depth in triplicates. Therefore, three 25 ml glass vials were filled bubble free directly after
CTD-rosette recovery and closed with butyl rubber stoppers. Samples were killed with saturated
mercury chloride solution and stored at room temperature until further treatment.
Concentrations of dissolved methane ($CH_4$) were determined by headspace gas chromatography as
described in Bange et al. (2010). Calibration for $CH_4$ was done by a two-point calibration with known
methane concentrations before the measurement of headspace gas samples, resulting in an error of
< 5 %.
Water samples for chlorophyll concentration were taken by transferring the complete water volume
(from 25 m water depth) from one water sampler into a 4.5 L Nalgene bottle, from which then
approximately 0.7-1 L (depending on the plankton content) were filtrated back in the GEOMAR
laboratory using GF/F filter (Whatman, 25 mm diameter, 8 µM pores size). Dissolved chlorophyll a
concentrations were determined using the fluorometric method by Welschmeyer (1994) with an
error < 10 %.
**2.4 Sediment porewater geochemistry**
Porewater was extracted from sediment within 24 hours after core retrieval using nitrogen ($N_2$) pre-
flushed rhizons (0.2 µm, Rhizosphere Research Products, Seeberg-Elverfeldt et al., 2005). In MUC
cores, rhizons were inserted into the sediment in 2 cm intervals through pre-drilled holes in the core
liner. In the gravity core, rhizons were inserted into the sediment in 30 cm intervals directly after
retrieval.
Extracted porewater from MUC and gravity cores was immediately analyzed for sulfide using
standardized photometric methods (Grasshoff et al., 1999).
Sulfate concentrations were determined using ion chromatography (Methrom 761). Analytical
precision was < 1 % based on repeated analysis of IAPSO seawater standards (dilution series) with an





absolute detection limit of 1 μM corresponding to a detection limit of 30 μM for the undiluted
sample.
For analysis of dissolved inorganic carbon (DIC), 1.8 ml of porewater was transferred into a 2 ml glass
vial, fixed with 10 μl saturated $HgCL_2$ solution and crimp sealed. DIC concentration was determined
as $CO_2$ with a multi N/C 2100 analyzer (Analytik Jena) following the manufacturer's instructions.
Therefore, the sample was acidified with phosphoric acid and the outgassing $CO_2$ was measured. The
detection limit was 20 μM with a precision of 2-3 %.

### 2.5 Sediment methane concentrations

In March 2013, June 2013 and March 2014, one MUC core was sliced in 1 cm intervals until 6 cmbsf,
followed by 2 cm intervals until the end of the core. At the other sampling months, the MUC core
was sliced in 1 cm intervals until 6 cmbsf, followed by 2 cm intervals until 10 cmbsf and 5 cm intervals
until the end of the core.
Per sediment depth (in MUC and gravity cores), 2 $cm^{-3}$ of sediment were transferred into a 10 ml-
glass vial containing 5 ml NaOH (2.5 %) for determination of sediment methane concentration per
volume of sediment. The vial was quickly closed with a butyl septum, crimp-sealed and shaken
thoroughly. The vials were stored upside down at room temperature until measurement via gas
chromatography. Therefore, 100 μl of headspace was removed from the gas vials and injected into a
Shimadzu gas chromatograph (GC-2014) equipped with a packed Haysep-D column and a flame
ionization detector. The column temperature was 80°C and the helium flow was set to 12 ml $min^{-1}$.
$CH_4$ concentrations were calibrated against $CH_4$ standards (Scotty gases). The detection limit was 0.1
ppm with a precision of 2 %.

### 2.6 Sediment solid phase geochemistry

Following the sampling for $CH_4$, the same cores described under section 2.5 were used for the
determination of the sediment solid phase geochemistry, i.e. porosity, particulate organic carbon
(POC) and particulate organic nitrogen (PON).
Sediment porosity of each sampled sediment section was determined by the weight difference of 5
$cm^{-3}$ wet sediment after freeze-drying for 24 hours. Dried sediment samples were then used for
analysis of particulate organic carbon (POC) and particulate organic nitrogen (PON) with a Carlo-Erba
element analyzer (NA 1500). The detection limit for C and N analysis was < 0.1 dry weight percent (%)
with a precision of < 2 %.

### 2.7 Sediment methanogenesis

### 2.7.1 Methanogenesis in MUC cores

At each sampling month, three MUC cores were sliced in 1 cm intervals until 6 cmbsf, in 2 cm
intervals until 10 cmbsf, and in 5 cm intervals until the bottom of the core. Every sediment layer was




transferred to a separate beaker and quickly homogenized before sub-sampling. The exposure time
with air, i.e. oxygen, was kept to a minimum. Sediment layers were then sampled for determination
of net methanogenesis (defined as the sum of total methane production and consumption, including
all available methanogenic substrates in the sediment), hydrogenotrophic methanogenesis
(methanogenesis based on the substrates $CO_2/H_2$), and potential methanogenesis (methanogenesis
at ideal conditions, i.e. no lack of nutrients) as described in the following sections.

### *Net methanogenesis*

Net methanogenesis was determined with sediment slurry experiments by measuring the headspace
methane concentration over time. Per sediment layer, triplicates of 5 cm$^{-3}$ of sediment were
transferred into $N_2$-flushed sterile glass vials (30 ml) and mixed with 5 ml filtered bottom water. The
slurry was repeatedly flushed with $N_2$ to remove residual methane and to ensure complete anoxia.
Slurries were incubated in the dark at in-situ temperature, which varied at each sampling date (Table
1). Headspace samples (0.1 ml) were taken out every 3-4 days over a time period of 4 weeks and
analyzed on a Shimadzu GC-2104 gas chromatograph (see Sect. 2.5). Net methanogenesis rates were
determined by the linear increase of the methane concentration over time (minimum of 6 time
points).

### *Hydrogenotrophic methanogenesis*

To determine hydrogenotrophic methanogenesis, radioactive sodium bicarbonate (NaH$^{14}$CO$_3$) was
added to the sediment.
Per sediment layer, sediment was sampled in triplicates with glass tubes (5 mL) which were closed
with butyl rubber stoppers on both ends according to (Treude et al. 2005). Through the stopper,
NaH$^{14}$CO$_3$ (dissolved in water, injection volume 6 µl, activity 222 kBq, specific activity = 1.85-2.22
GBq/mmol) was injected into each sample and incubated for three days in the dark at in-situ
temperature (Table 1). To stop bacterial activity, sediment was transferred into 50 ml glass-vials filled
with 20 ml sodium hydroxide (2.5 % w/w), closed quickly with rubber stoppers and shaken
thoroughly. Five controls were produced from various sediment depths by injecting the radiotracer
directly into the NaOH with sediment.
The production of $^{14}$C-methane was determined with the slightly modified method by Treude et al.,
(2005) used for the determination of anaerobic oxidation of methane. The method was identical,
except no unlabeled methane was determined by gas chromatography. Instead, DIC values were
used to calculate hydrogenotrophic methane production.

### *Potential methanogenesis in manipulated experiments*

To examine the interaction between sulfate reduction and methanogenesis, inhibition and
stimulation experiments were carried out. Therefore, every other sediment layer was sampled



resulting in the following examined six sediment layers: 0-1 cm, 2-3 cm, 4-5 cm, 6-8 cm, 10-15 cm
and 20-25 cm. From each layer, sediment slurries were prepared by mixing 5 ml sediment in a 1:1
ratio with adapted artificial seawater medium (salinity 24, Widdel & Bak, 1992) in $N_2$-flushed, sterile
glass vials before further manipulations.
In total, four different treatments, each in triplicates, were prepared per depth: 1) with sulfate
addition (17 mM), 2) with sulfate (17 mM) and molybdate (22 mM) addition, 3) with sulfate (17 mM)
and 2-bromoethane-sulfonate (BES, 60 mM) addition, and 4) with sulfate (17 mM) and methanol (10
mM) addition. From here on, the following names are used to describe the different treatments,
respectively: 1) control treatment, 2) molybdate treatment, 3) BES treatment, and 4) methanol
treatment. Control treatments feature the natural sulfate concentrations occurring in surface
sediments of the sampling site. Molybdate was used as an enzymatic inhibitor for sulfate reduction
(Oremland & Capone, 1988) and BES was used as an inhibitor for methanogenic archaea (Hoehler et
al., 1994). Methanol is a known non-competitive substrate, which is used by methanogens but not by
sulfate reducers (Oremland & Polcin, 1982), thus it is suitable to examine non-competitive
methanogenesis. Treatments were incubated at the respective in-situ temperature (Table 1) in the
dark.

### Potential methylotrophic methanogenesis from methanol using stable isotope probing

One additional experiment was conducted with sediments from September 2014 by adding $^{13}$C-
labelled methanol to investigate the production of $^{13}$C-labelled methane. Three cores were stored at
1°C after the September 2014 cruise until further processing ~ 3.5 months later. The low storage
temperature and the fast oxygen consumption in the enclosed supernatant water (i.e., exclusion of
bioturbation by macrofauna) led to slowed microbial activity and preserved the sediments for
potential methanogenesis measurements.
Sediment cores were sliced in 2 cm intervals and the upper 0-2 cmbsf sediment layer of all three
cores was combined in a beaker and homogenized. Then, sediment slurries were prepared by mixing
5 cm$^{-3}$ of sediment with 5 ml of artificial seawater medium in $N_2$-flushed, sterile glass vials (30 ml).
Then, methanol was added to the slurry with a final concentration of 10 mM (see Sect. 2.7.3), but
this time the methanol was enriched with $^{13}$C-labelled methanol in a ratio of 1:1000 between $^{13}$C-
labelled (99.9 % $^{13}$C) and non-labelled methanol mostly consisting of $^{12}$C (manufacturer: Roth). In
total, 54 vials were prepared for nine different sampling time points during a total incubation time of
37 days. All vials were incubated at 13°C (in situ temperature in September 2014) in the dark. At each
sampling point, six vials were stopped: one set of triplicates were used for headspace methane and
carbon dioxide determination and a second set of triplicates were used for porewater analysis.
Headspace methane and carbon dioxide concentrations (volume 100 µl) were determined on a
Shimadzu gas chromatograph (GC-2014) equipped with a packed Haysep-D column a flame ionization





detector and a methanizer. The methanizer (reduced nickel) reduces carbon dioxide with hydrogen
to methane at a temperature of 400°C. The column temperature was 80°C and the helium flow was
set to 12 ml min$^{-1}$. Methane concentrations (including reduced $CO_2$) were calibrated against methane
standards (Scotty gases). The detection limit was 0.1 ppm with a precision of 2 %.
Analyses of $^{13}C/^{12}C$-ratios of methane and carbon dioxide were conducted after headspace
concentration measurements by using a continuous flow combustion gas chromatograph (Trace
Ultra, Thermo Scientific), which was coupled to an isotope ratio mass spectrometer (MAT253,
Thermo Scientific).  The isotope ratios of methane and carbon dioxide given in the common delta-
notation ($\delta$ $^{13}C$ in permill) are reported relative to Vienna Pee Dee Belemnite (VPDB) standard.
Isotope precision was +/- 0.5 ‰, when measuring near the detection limit of 10 ppm.
For porewater analysis of methanol concentration and isotope composition, each sediment slurry of
the triplicates was transferred into argon-flushed 15 ml centrifuge tubes and centrifuged for 6
minutes at 4500 rpm. Then 1 ml filtered (0.2 μm) porewater was transferred into $N_2$-flushed 2 ml
glass vials for methanol analysis, crimp sealed and immediately frozen at -20 °C. Methanol
concentrations and isotope composition were determined via high performance liquid
chromatography-ion ratio mass spectrometry (HPLC-IRMS, Thermo Fisher Scientific) at the MPI
Marburg. The detection limit was 50 μM with a precision of 0.3‰.

### 315    2.7.2 Methanogenesis in the gravity core

Ex situ hydrogenotrophic methanogesis was determined in a gravity core taken September 2013. The
pathway is thought to be the main methanogenic pathway in the deep sediment layers (below
sulfate penetration) in Eckernförde Bay (Whiticar, 2002). Hydrogenotrophic methanogenesis was
determined using $^{14}C$-bicarbonate. At every sampled sediment depth (12 depths in 30 cm intervals),
triplicate glass tubes (5 mL) were inserted directly into the sediment. Tubes were filled bubble-free
with sediment and closed with butyl rubber stoppers on both ends according to (Treude et al. 2005).
Methods following sampling were identical as described in 2.7.2.

### 323    2.8 Molecular analysis

In September 2014, additional samples were prepared for the methanol treatment of the 0-1 cmbsf
horizon during the potential methanogenesis experiment described in 2.7.3 to detect and quantify
the presence of methanogens in the sediment. Therefore, additional 15 vials were prepared with
addition of methanol as described in 2.7.3 for five different time points (day 1 (= $t_0$), day 8, day 16,
day 22, and day 36) and stopped at each time point by transferring sediment from the triplicate
slurries into whirl-packs (Nasco), which then were immediately frozen at -20°C. DNA was extracted
from ~500 mg of sediment using the FastDNA® SPIN Kit for Soil (Biomedical). Quantitative real-time
polymerase chain reaction (qPCR) technique using TaqMan probes and TaqMan chemistry (Life



Technologies) was used for the detection of methanogens on a ViiA7 qPCR machine (Life
Technologies). Primer and Probe sets as originally published by Yu et al. (2005) were applied to
quantify the orders *Methanobacteriales, Methanosarcinales* and *Methanomicrobiales* along with the
two families *Methanosarcinaceae* and *Methanosaetaceae* within the order *Methanosarcinales.* In
addition, a universal primer set for detection of the domain *Archaea* was used (Yu et al. 2005).
Absolut quantification of the 16S rDNA from the groups mentioned above was performed with
standard dilution series. The standard concentration reached from $10^8$ to $10^1$ copies per µL.
Quantification of the standards and samples was performed in duplicates. Reaction was performed in
a final volume of 12.5 µL containing 0.5 µL of each Primer (10pmol µL$^{-1}$, MWG), 0.25 µL of the
respective probe (10 pmol µL$^{-1}$, Life Technologies), 4 µL $H_2O$ (Roth), 6.25 µL TaqMan Universal Master
Mix II (Life Technologies) and 1 µL of sample or standard. Cycling conditions started with initial
denaturation and activation step for 10 min at 95°C, followed by 45 cycles of 95 °C for 15 sec, 56°C
for 30 sec and 60°C for 60 sec. Non-template controls were run in duplicates with water instead of
DNA for all primer and probe sets, and remained without any detectable signal after 45 cycles.
### 2.9 Statistical Analysis
To determine possible environmental controlling parameters on surface methanogenesis, a Principle
Component Analysis (PCA) was applied according to the approach described in Gier et al.( 2016).
Prior to PCA, the dataset was transformed into ranks to assure the same data dimension.
In total, two PCAs were conducted. The first PCA was used to test the relation of parameters in the
surface sediment (integrated methanogenesis (0-5 cm, mmol m$^{-2}$ d$^{-1}$), POC content (average value
from 0-5 cmbsf, wt %), C/N (average value from 0-5 cmbsf, molar) and the bottom water (25 m water
depth) (oxygen (µM), temperature (°C), salinity (PSU), chlorophyll (µg L$^{-1}$), methane (nM)). The
second PCA was applied on depth profiles of sediment surface methanogenesis (nmol cm$^{-3}$ d$^{-1}$),
sediment depth (cm), sediment POC content (wt%), sediment C/N ratio (molar), and sampling month
(one value per depth profile at a specific month, the later in the year the higher the value).
For each PCA, biplots were produced to view data from different angles and to graphically determine
a potential positive, negative or zero correlation between methanogenesis rates and the tested
variables.
## 3. Results
### 3.1 Water column parameters
From March 2013 to September 2014, the water column had a pronounced temporal and spatial
variability of temperature, salinity, and oxygen (Fig. 1 and 2). In 2013, temperature of the upper
water column increased from March (1°C) to September (16°C), but decreased again in November





(11°C). The temperature of the lower water column increased from March 2013 (2˚C) to November
2013 (12˚C). In 2014, lowest temperatures of the upper and lower water column were reached in
March (4°C). Warmer temperatures of the upper water column were observed in June and
September (around 17°C), while the lower water column peaked in September (13˚C).
Salinity increased over time during 2013, showing the highest salinity of the upper and lower water
column in November (18 and 23 PSU, respectively). In 2014, salinity of the upper water column was
highest in March and September (both 17 PSU), and lowest in June (13 PSU). The salinity of the lower
water column increased from March 2014 (21 PSU) to September 2014 (25 PSU).
In both years, June and September showed the most pronounced vertical gradient of temperature
and salinity, featuring a pycnocline at around ~14 m water depth.
Summer stratification was also seen in the $O_2$ profiles, which showed $O_2$ depleted conditions ($O_2 <$
µM) in the lower water column from June to September in both years, reaching concentrations
below 1- 2 µM (detection limit of CTD sensor) in September of both years (Fig. 1 and 2). The water
column was completely ventilated, i.e. homogenized, in March of both years with $O_2$ concentrations
of 300-400 µM down to the sea floor at about 28 m.

3.2 Sediment geochemistry in MUC cores
Sediment porewater and solid phase geochemistry results for the years 2013 and 2014 are shown in
Fig. 1 and 2, respectively.
Sulfate concentrations at the sediment surface ranged between 15-20 mM. Concentration decreased
with depth at all sampling months but was never fully depleted until the bottom of the core (18-29
cmbsf, between 2 and 7 mM sulfate). November 2013 showed the strongest decrease from ~20 mM
at the top to ~2 mM at the bottom of the core (27 cmbsf).
Opposite to sulfate, methane concentration increased with sediment depth in all sampling months
(Fig. 1 and 2). Over the course of a year (i.e. March to November in 2013, and March to September in
2014), maximum methane concentration increased, reaching the highest concentration in November
2013 (~1 mM at 26 cmbsf) and September 2014 (0.2 mM at 23 cmbsf), respectively. Simultaneously,
methane profiles became steeper, revealing higher methane concentrations at shallower sediment
depth late in the year. Magnitudes of methane concentrations were similar in the respective months
of 2013 and 2014.
In all sampling months, sulfide concentration increased with sediment depth (Fig. 1 and 2). Similar to
methane, sulfide profiles revealed higher sulfide concentrations at shallower sediment depth
together with higher peak concentrations over the course the sampled months in each sampling
year. Accordingly, November 2013 (10.5 mM at 15 cmbsf) and in September 2014 (2.8 mM at 15
cmbsf) revealed the highest sulfide concentrations, respectively. September 2014 was the only

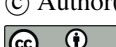



sampling month showing a pronounced decrease in sulfide concentration from 15 cmbsf to 21 cmbsf
of over 50 %.
DIC concentrations increased with increasing sediment depth at all sampling months. Concomitant
with highest sulfide concentrations, highest DIC concentration was detected in November 2013 (26
mM at 27 cmbsf). At the surface, DIC concentrations ranged between 2-3 mM at all sampling
months. In June of both years, DIC concentrations were lowest at the deepest sampled depth
compared to the other sampling months (16 mM in 2013, 13 mM in 2014).
At all sampling months, POC profiles scattered around 5 ± 0.9 wt % with depth. Only in November
2013, June 2014 and September 2014, POC content exceeded 5 wt % in the upper 0-1 cmbsf (5.9, 5.2
and 5.3 wt %, respectively) with the highest POC content in November 2013. Also in November 2013,
surface C/N ratio was lowest of all sampling months (8.6). In general, C/N ratio increased with depth
in both years with values around 9 at the surface and values around 10-11 at the deepest sampled
sediment depths.

### 3.3 Sediment geochemistry in gravity cores

Results from sediment porewater and solid phase geochemistry in the gravity core from September
2013 are shown in Fig. 3. Please note that the sediment depth of the gravity core was corrected by
comparing the sulfate concentrations at 0 cmbsf in the gravity core with the corresponding sulfate
concentration and depth in the MUC core from September 2013 (Fig. 1). The soft surface sediment is
often lost during the gravity coring procedure. Through this correction the topmost layer of the
gravity core was set at a depth of 14 cmbsf.
Porewater sulfate concentration in the gravity core decreased with depth (i.e. below 0.1 mM at 107
cmbsf) and stayed below 0.1 mM until 324 cmbsf. Sulfate increased slightly (1.9 mM) at the bottom
of the core (345 cmbsf). In concert with sulfate, also methane, sulfide, DIC, POC and C/N profiles
showed distinct alteration in the profile at 345 cmbsf (see below, Fig. 3). As fluid seepage has not
been observed at the Boknis Eck station (Schlüter et al., 2000), these alterations could either indicate
a change in sediment properties or result from a sampling artifact from the penetration of seawater
through the core catcher into the deepest sediment layer. The latter process is, however, not
expected to considerably affect sediment solid phase properties (POC and C/N), and we therefore
dismissed this hypothesis.
Methane concentration increased steeply with depth reaching a maximum of 4.8 mM at 76 cmbsf.
Concentration stayed around 4.7 mM until 262 cmbsf, followed by a slight decrease until 324 cmbsf
(2.8 mM). From 324 cmbsf to 345 cmbsf methane increased again (3.4 mM).
Both sulfide and DIC concentrations increased with depth, showing a maximum at 45 cmbsf (~ 5mM)
and 345 cmbsf (~ 1mM), respectively. While sulfide decreased after 45 cmbsf to a minimum of ~ 300
μM at 324 cmbsf, it slightly increased again to ~1 mM at 345 cmbsf. In accordance, DIC



concentrations showed a distinct decrease between 324 cmbsf to 345 cmbsf (from 45 mM to 39
mM).
While POC concentrations varied around 5 wt % throughout the core, C/N ratio slightly increased
with depth, revealing the lowest ratio at the surface (~3) and the highest ratio at the bottom of the
core (~13). However, both POC and C/N showed a distinct increase from 324 cmbsf to 345 cmbsf.

3.4 Methanogenesis activity in MUC cores
*3.4.1 Net methanogenesis*
Net methanogenesis activity was detected throughout the cores at all sampling months (Fig. 1 and 2).
Activity measured in MUC cores increased over the course of the year in 2013 and 2014 (that is:
March to November in 2013 and March to September in 2014) with lower rates mostly < 0.1 nmol
$cm^{-3}$ $d^{-1}$ in March and higher rates > 0.2 nmol $cm^{-3}$ $d^{-1}$ in November 2013 and September 2014,
respectively. In general, November 2013 revealed highest net methanogenesis rates (1.3 nmol $cm^{-3}$ $d^{-1}$
at 1-2 cmbsf). Peak rates were detected at the sediment surface (0-1 cmbsf) at all sampling months
except for September 2013 where the maximum rates were situated between 10-15 cmbsf. In
addition to the surface peaks, net methanogenesis showed subsurface (= below 1 cmbsf until 30
cmbsf) maxima at all sampling months, but with alternating depths (between 10 and 25 cmbsf).
Comparison of integrated net methanogenesis rates (0-25 cmbsf) revealed highest rates in
September and November 2013 and lowest rates in March 2014 (Fig. 4). A trend of increasing areal
net methanogenesis rates from March to September was observed in both years.
*3.4.2 Hydrogenotrophic methanogenesis*
Hydrogenotrophic methanogenesis activity determined by $^{14}$C-bicarbonate incubations of MUC cores
is shown in Fig. 1 and 2. In 2013, maximum activity ranged between 0.01-0.2 nmol $cm^{-3}$ $d^{-1}$, while in
2014 maxima ranged only between 0.01 and 0.05 nmol $cm^{-3}$ $d^{-1}$. In comparison, maximum
hydrogenotrophic methanogenesis was up to two orders of magnitude lower compared to net
methanogenesis. Only in March 2013 both activities reached a similar range.
Overall, hydrogenotrophic methanogenesis increased with depth in March, September, and
November 2013 and in March, June, and September 2014. In June 2013, activity decreased with
depth, showing the highest rates in the upper 0-5 cmbsf and the lowest at the deepest sampled
depth.
Concomitant with integrated net methanogenesis, integrated hydrogenotrophic methanogenesis
rates (0-25 cmbsf) were high in September 2013, with slightly higher rates in March 2013 (Fig. 4).
Lowest areal rates of hydrogenotrophic methanogenesis were seen in June of both years.





Hydrogenotrophic methanogenesis activity in the gravity core is shown in Fig. 3. Highest activity (~
0.7 nmol cm$^{-3}$ d$^{-1}$) was measured at 45 cmbsf and 138 cmbsf, followed by a decrease with increasing
sediment depth reaching 0.01 nmol cm$^{-3}$ d$^{-1}$ at the deepest sampled depth (345 cmbsf).
***3.4.3 Potential methanogenesis in manipulated experiments***
Potential methanogenesis rates in manipulated experiments included either the addition of
inhibitors (molybdate for inhibition of sulfate reduction or BES for inhibition of methanogenesis) or
the addition of a non-competitive substrate (methanol). Control treatments were run with neither
the addition of inhibitors nor the addition of methanol.
*Controls*. Potential methanogenesis activity in the control treatments was below 0.5 nmol cm$^{-3}$ d$^{-1}$
from March 2014 to September 2014 (Fig. 5). Only in November 2013, control rates exceeded 0.5
nmol cm$^{-3}$ d$^{-1}$ below 6 cmbsf. While rates increased with depth in November 2013 and June 2014,
they decreased with depth at the other two sampling months.
*Molybdate*. Peak potential methanogenesis rates in the molybdate treatments were found in the
uppermost sediment interval (0-1 cmbsf) at almost every sampling month with rates being 3-30
times higher compared to the control treatments (< 0.5 nmol cm$^{-3}$ d$^{-1}$). In November 2013, potential
methanogenesis showed two maxima (0-1 and 10-15 cmbsf). Highest measured rates were found in
September 2014 (~6 nmol cm$^{-3}$ d$^{-1}$), followed by November 2013 (~5 nmol cm$^{-3}$ d$^{-1}$).
*BES*. Profiles of potential methanogenesis in the BES treatments were similar to the controls mostly
in the lower range < 0.5 nmol cm$^{-3}$ d$^{-1}$. Only in November 2013 rates exceeded 0.5 nmol cm$^{-3}$ d$^{-1}$.
Rates increased with depth at all sampling months, except for September 2014, where highest rates
were found at the sediment surface (0-1 cmbsf).
*Methanol*. At all sampling months, potential rates in the methanol treatments were three orders of
magnitude higher compared to the control treatments (< 0.5 nmol cm$^{-3}$ d$^{-1}$). Except for November
2013, potential methanogenesis rates in the methanol treatments were highest in the upper 0-5
cmbsf and decreased with depth. In November 2013, highest rates were detected at the deepest
sampled depth (20-25 cmbsf).

***3.4.4 Potential methanogenesis determined from $^{13}$C-labelled methanol***
The concentration of methanol in the sediment decreased sharply in the first 2 weeks from ~8 mM at
day 1 to 0.5 mM at day 13 (Fig. 6). At day 17, methanol was below the detection limit. In the first 2
weeks, residual methanol was enriched with $^{13}$C, reaching ~200 ‰ at day 13.
Over the same time period, the concentration of methane increased from 2 ppmv at day 1 to ~
66,000 ppmv at day 17 and stayed around that value until the end of the total incubation time (until
day 37) (Fig. 6). The carbon isotopic signature of methane ($\delta^{13}C_{CH4}$) showed a clear enrichment of the
heavier isotope $^{13}$C (Table 3) from day 9 to 17 (no methane was detectable at day 1). After day 17,



$\delta^{13}C_{CH4}$ stayed around 13‰ until the end of the incubation. The concentration of $CO_2$ in the
headspace increased from ~8900 ppmv at day 1 to ~29,000 ppmv at day 20 and stayed around
30,000 ppmv until the end of the incubation (Fig. 6). Please note, that the major part of $CO_2$ was
dissolved in the porewater, thus the $CO_2$ concentration in the headspace does not show the total $CO_2$
concentration in the system. $CO_2$ in the headspace was enriched with $^{13}C$ during the first 2 weeks
(from -16.2 to -7.3 ‰) but then stayed around -11 ‰ until the end of the incubation.
### 3.5 Molecular analysis of benthic methanogens
In September 2014, additional samples were run during the methanol treatment (see Sect. 2.7.3) for
the detection of benthic methanogens via qPCR. The qPCR results are shown in Fig. 7. For a better
comparison, the microbial abundances are plotted together with the sediment methane
concentrations from the methanol treatment, from which the rate calculation for the methanol-
methanogenesis at 0-1 cmbsf was done (shown in Fig. 5).
Methane concentrations increased over time revealing a slow increase in the first ~10 days, followed
by a steep increase between day 13 and day 20 and ending in a stationary phase.
A similar increase was seen in the abundance of total and methanogenic archaea. Total archaea
abundances increased sharply in the second week of the incubation reaching a maximum at day 16
(~5000 $*10^6$ copies $g^{-1}$) and stayed around 3000 $*10^6$-4000 $*10^6$ copies $g^{-1}$ over the course of the
incubation. Similarly, methanogenic archaea, namely the order *Methanosarcinales* and within this
order the family *Methanosarcinaceae*, showed a sharp increase in the first 2 weeks as well with the
highest abundances at day 16 (~6* $10^8$ copies $g^{-1}$ and ~1*$10^6$ copies $g^{-1}$, respectively). Until the end of
the incubation, the abundances of *Methanosarcinales* and *Methanosarcinaceae* decreased to about a
third of their maximum abundances (~2*$10^8$ copies $g^{-1}$ and ~0.4*$10^6$ copies $g^{-1}$, respectively).
### 3.6 Statistical Analysis
The PCA of integrated surface methanogenesis (0-5 cmbsf) (Fig.10) showed a strong positive
correlation with bottom water temperature (Fig. 9a), bottom water salinity (Fig. 9a), and surface
sediment POC content (Fig. 9c). Further, a positive correlation with bottom water methane and a
weak positive correlation with surface sediment C/N was detected (Fig. 9b). A strong negative
correlation was found with bottom water oxygen concentration (Fig. 9b). No correlation was found
with bottom water chlorophyll.
The PCA of methanogenesis depth profiles showed weak positive correlations with sediment depth
(Fig. 10a) and C/N (Fig. 10b), and showed negative correlations with POC (Fig. 10a).





## 4. Discussion

### 4.1 Methanogenesis in the sulfate-reducing zone

On the basis of the results presented in Fig. 1 and 2, it is evident that methanogenesis and sulfate reduction were concurrently active in the surface sediments (0-30 cmbsf) at Boknis Eck. Even though sulfate reduction rates were not measured directly, the decrease in sulfate concentrations with a concomitant increase in sulfide within the upper 30 cmbsf indicate that sulfate reduction was active (Fig. 1 and 2). Several earlier studies in Eckernförde Bay sediments confirmed the dominance of sulfate reduction in the surface sediment, which revealed an activity of 100-10000 nmol $cm^{-3}$ $d^{-1}$ in the upper 25 cmbsf (Treude et al., 2005a; Bertics et al., 2013; Dale et al., 2013). Microbial fermentation of organic matter was probably high in the organic-rich sediments of Eckernförde Bay (POC contents of around 5 %, Fig. 1 and 2), providing high substrate availability and variety for methanogenesis.

The results of this study further identified methylotrophy to be an important non-competitive methanogenic pathway in the sulfate-reducing zone. The pathway utilizes alternative substrates, such as methanol, to avoid competition with sulfate reducers for $H_2$ and acetate. The relevance of methylotrophic methanogenesis in the sulfate-reducing zone was supported by the following observations: 1) Hydrogenotrophic methanogenesis was up to two orders of magnitude lower than net methanogenesis (Fig. 1 and 2), 2) methanogenesis increased when sulfate reduction was inhibited (Fig. 5), 3) addition of BES did not result in the inhibition of methanogenesis (Fig. 6), 4) addition of methanol increased potential methanogenesis rates up to three orders of magnitude (Fig. 6), 5) methylotrophic methanogens of the order *Methanosarcinales* were detected in the methanol-treatment (Fig. 7), and 6) stable isotope probing revealed highly $^{13}C$-enriched methane produced from $^{13}C$-labelled methanol (Fig. 6). In the following chapters, these arguments will be discussed in more detail.

### 4.1.1 Hydrogenotrophic methanogenesis

We demonstrated that hydrogenotrophic methanogenesis was insufficient to explain the observed net methanogenesis. The only exemption was March 2013, where rates of hydrogenotrophic methanogenesis exceeded net methanogenesis in discrete depths (5-6 cmbsf and 25-30 cmbsf). It is possible that additional carbon sources led to increased local fermentation processes, for instance from the deposition of macro algae detritus, which is produced during winter storms and can be transported into deeper sediment layers by bioturbation, where it is digested and released as fecal pellets (Meyer-Reil, 1983; Bertics et al., 2013). Such additional carbon sources from fresh material could lead to the local accumulation of excess hydrogen through fermentation and reduce the



competition for $H_2$ between sulfate reducers and methanogens (Treude et al., 2009). C/N ratios in
March 2013 were more scattered compared to other months in 2013 and 2014, indicating the
transport of labile material into the sediment. Eckernförde Bay sediments are known for bioturbation
especially during early spring by mollusks and polycheates (D'Andrea et al., 1996; Orsi et al., 1996;
Bertics et al., 2013; Dale et al., 2013), and mollusk shells were observed even at depth of ~ 20 cmbsf
during sampling in the present study (personal observation).
Hydrogenotrophic methanogenesis was also detected in the gravity core in September 2013.
Maximum hydrogenotrophic rates were found at 45 cmbsf and 138 cmbsf, indicating a higher usage
of $CO_2$ and $H_2$ at depths > 40 cmbsf, where sulfate was depleted and thus the competition between
sulfate reducers and methanogens was relieved.
### 4.1.2 Inhibition of sulfate reducers
The competition between methanogens and sulfate reducers within the upper 30 cmbsf led to the
predominant utilization of non-competitive substrates by methanogenesis, as indicated by low
hydrogenotrophic methanogenesis rates (see discussion above). After the addition of the sulfate-
reducer inhibitor molybdate, competitive substrates ($H_2/CO_2$ and acetate (Oremland & Polcin, 1982;
King et al., 1983) were available for methanogenesis as indicated by the increase (up to 30 times) in
potential activity (Fig. 5 and 6). Notably, highest rates in the molybdate treatment were measured at
the shallowest sediment depth at most sampling months (except November 2013), pointing towards
the strongest competition between sulfate reducers and methanogens directly at the top 0-1 cmbsf,
which is confirmed by sulfate reduction maxima found at 0-1 cmbsf in earlier studies (Bertics et al.
2013; Treude et al. 2005).
### 4.1.3 Inhibition of methanogenesis by BES
Addition of BES did not result in the expected inhibition of potential methanogenesis; instead rates
were in the same range as the control treatment (Fig. 6). Either the inhibition of BES was incomplete,
or the methanogens were insensitive to BES (Hoehler et al., 1994; Smith & Mah, 1981; Santoro &
Konisky, 1987). However, the BES concentration used in the present study (60 mM) has been shown
to result in successful inhibition of methanogens in previous studies (Hoehler et al., 1994). Therefore,
the presence of methanogens that are insensitive BES was more likely. Insensitivity to BES would
support the hypothesis that methanogenesis in the sulfate reduction zone is mainly driven via the
methylotrophic pathway, as BES resistance was shown in *Methanosarcina* mutants in earlier studies
(Smith & Mah, 1981; Santoro & Konisky, 1987), a genus which we successfully detected in our
samples (for more details see Sect. 4.1.5), and which is known for mediating the methylotrophic
pathway (Keltjens & Vogels, 1993).





### 4.1.4 Methanol addition
High potential methanogenesis rates observed after the addition of the non-competitive substrate
methanol leads to the assumption that non-competitive substrates relieve the competition between
methanogens and sulfate reducers in surface sediments of Eckernförde Bay. Except for November
2013, highest rates in the methanol-treatment were detected in the upper 0-5 cmbsf and decreased
with depth (Fig. 5). Highest methanogenesis rates in the upper 0-5 cmbsf of the methanol-treatment
can be interpreted as follows: (1) The amount of non-competitive substrates including methanol was
most likely highest at the sediment surface, as those substrates are derived from fresh organic
matter, such as pectin or betaine and dimethylpropiothetin (both osmoprotectants) (Zinder, 1993).
(2) Sulfate reduction is most dominant in the 0-5 cmbsf (Treude et al., 2005a; Bertics et al., 2013),
which probably leads prevalent methanogens to be more adapted to the usage if non-competitive
substrates.
It should be noted that even though methanogenesis rates were calculated assuming a linear
increase in methane concentration over the entire incubation to make a better comparison between
different treatments, the methanol treatments generally showed a delayed response in methane
development (Supplement, Fig. S1). A similar delay was observed in organic-rich surface sediments
sampled off Peru and was explained by the predominant use of alternative non-competitive
substrates such as methylated sulfides (e.g. dimethyl sulfide or methanethiol (Maltby et al., 2016)). In
the marine environment, dimethyl sulfide mainly originate from the algae osmoregulatory compound
dimethylsulfoniopropionate (DMSP) (Van Der Maarel & Hansen, 1997), which could have
accumulated in Eckernförde Bay sediments, due to intense sedimentation of algae blooms (Bange et
al., 2011). Certain *Methanosarcina* species have been shown to use DMS as a substrate (Sieburth et
al., 1993; Van Der Maarel & Hansen, 1997), a genus, which has been detected in our samples (see
more details under Sect. 4.1.5).
Additionally, there are hints that methylated sulfur compounds may be generated through
nucleophilic attack by sulfide on the methyl groups in the sedimentary organic matter (Mitterer,
2010). As shown in the present study, sulfide was an abundant species in the surface sediment (up to
mM levels) (Fig. 1 and 2).
### 4.1.5 Presence of methylotrophic methanogens
Simultaneously with the increase in methane concentration after methanol addition in the surface
layer (0-1 cmbsf) in September 2014, the DNA counts for the order *Methanosarcinales* and the family
*Methanosarcinaceae* within the order *Methanosarcinales* increased 102 to $10^6$ times, respectively,
compared to the respective DNA abundances at the start of the incubation (Fig. 7). The successful
enrichment of *Methanosarcinaceae* indicates that this family is present in the natural environment
and thus could in part be responsible for the observed surface methanogenesis. As the members of



the family *Methanosarcinaceae* are known for utilization of methylated substrates (Boone et al.,
1993), our hypothesis for the predominant usage of non-competitive substrates is supported. The
delay in growth of *Methanosarcinales* and *Methanosarcineceae*, however, also hints towards the
predominant usage of other non-competitive substrates besides methanol (see also Sect. 4.1.4).
**4.1.6 Stable-isotope experiment**
Samples taken in September 2014 for the labeling experiment ([13]C-enriched methanol, initial isotopic
signature: +26 ‰) showed that methanol was completely consumed after 17 days and converted to
methane and $CO_2$, as both revealed a concomitant enrichment in [13]C. The production of both
methane and $CO_2$ from methanol has been shown previously in different strains of methylotrophic
methanogens (Penger et al., 2012). As mentioned earlier, the major part of $CO_2$ was dissolved in the
porewater, which was not determined isotopically in this study, which is why we neglect the $CO_2$
development in the following.
Fractionation factors of methylotrophic methanogenesis from methanol to methane have been
found to be 1.07-1.08 (Heyer et al., 1976; Krzycki et al., 1987). This fractionation leads to a
progressive enrichment of [13]C in the residual methanol until all methanol is consumed. Accordingly,
methanol was enriched in [13]C in the first 13 days, as the consumption of [12]C-methanol was preferred
by the microbes. The fast conversion of methanol to methane can only be explained by the presence
of methylotrophic methanogens (e.g. members of the family *Methanosarcinaceae*, which is known
for the methylotrophic pathway (Keltjens & Vogels, 1993). Please note, however, that the storage of
the cores (3.5 months) prior to sampling could have led to shifts in the microbial community and thus
might not reflect in-situ conditions of the original microbial community in September 2014. The delay
in methane production also seen in the stable isotope experiment was, however, only slightly
different (methane developed earlier, between day 8 and 12, data not shown) from the non-labeled
methanol treatment (between day 10 to 16, Fig. S1), which leads us to the assumption that the
storage time at 1˚C did not dramatically affect the methanogen community. Similar, in a previous
study with arctic sediments, addition of substrates had no stimulatory effect on the rate of
methanogenesis or on the methanogen community structure at low temperatures (5˚C, (Blake et al.,

2015).

**4.2 Environmental control of surface methanogenesis**
Surface methanogenesis in Eckernförde Bay sediments showed variations throughout the sampling
period, which may be influenced by variable environmental factors such as temperature, salinity,
oxygen, and organic carbon. In the following, we will discuss the potential impact of those factors on
the magnitude and distribution of surface methanogenesis.
***4.2.1 Temperature***



During the sampling period, bottom water temperatures increased over the course of the year from
late winter (March, 3-4 °C) to autumn (November, 12°C, Fig. 1 and 2). The PCA revealed a strong
positive correlation between bottom water temperature and integrated surface methanogenesis (0-5
cmbsf). A temperature experiment conducted with sediment from ~75 cmbsf in September 2014
within a parallel study revealed a mesophilic temperature optimum of methanogenesis (20 °C, data
not shown). Whether methanogenesis in surface sediments (0-30 cm) has the same physiology
remains speculative. However, AOM organisms, which are closely related to methanogens (Knittel &
Boetius, 2009), studied in surface sediments from the same site were confirmed to have a mesophilic
physiology, too (Treude et al. 2005).

### *4.2.2 Salinity and oxygen*
From March 2013 to November 2013, and from March 2014 to September 2014, salinity increased in
the bottom-near water (25 m) from 19 to 23 PSU and from 22 to 25 PSU (Fig. 1 and 2), respectively,
due the pronounced summer stratification in the water column between saline North Sea water and
less saline Baltic Sea water (Bange et al., 2011). The PCA detected a strong positive correlation
between integrated surface methanogenesis (0-5 cmbsf) and salinity in the bottom-near water (Fig.
9a). This correlation can hardly be explained by salinity alone, as methanogens feature a broad
salinity range from freshwater to hypersaline (Zinder, 1993). Even more, methanogenesis often
decreases with increasing salinity (Pattnaik et al., 2000), due to the concurrent increase of sulfate,
enabling sulfate-reducing bacteria to degrade organic matter prior to hydrogenotrophic and
acetoclastic methanogens  (Oremland & Polcin, 1982). In fact, we found steep sulfate and sulfide
profiles at times of high salinity, indicating the presence of extensive sulfate reduction activity at the
sediment-water interface (Fig. 1 and 2). We therefore interpret positive correlation of
methanogenesis with salinity as an indirect indicator for a positive correlation with water column
stratification and hypoxia development. Accordingly, the PCA revealed a strong negative correlation
between oxygen concentration close to the seafloor and surface methanogenesis. In September
2014 bottom water levels probably reached zero levels as sulfide was detected in the bottom-near
water (25 m) 6 days after our sampling (H. Bange, pers. comm.). Hypoxia or anoxia in the bottom-
near water and the correlated absence of bioturbating and bioirrigating macrofauna (Dale et al.,
2013; Bertics et al., 2013) likely increased the habitable zone of methanogens close to the sediment-
water interface. Oxygen is an important factor controlling methanogenesis, as benthic methane is
mostly produced under strictly anoxic, highly reducing (< -200 mV) conditions (Oremland, 1988;
Zinder, 1993).

### *4.2.4 Particulate organic carbon*



The supply of particulate organic carbon (POC) is one of the most important factors controlling benthic heterotrophic processes, as it determines substrate availability and variety (Jørgensen, 2006). In Eckernförde Bay, the organic material reaching the sediment floor originates mainly from phytoplankton blooms in spring, summer and autumn (Bange et al., 2011). It has been estimated that > 50 % in spring (February/March), > 25 % in summer (July/August) and > 75 % in autumn (September/October) of these blooms is reaching the seafloor (Smetacek et al., 1984), resulting in a overall high organic carbon content of the sediment (5 wt %), which leads to high benthic microbial degradation rates including sulfate reduction  and methanogenesis (Whiticar, 2002; Treude et al., 2005a; Bertics et al., 2013). Previous studies revealed that high organic matter availability can relieve competition between sulfate reducers and methanogens in sulfate-containing, marine sediments (Oremland et al., 1982; Holmer & Kristensen, 1994; Treude et al., 2009; Maltby et al., 2016).

To determine the effect of POC concentration and C/N ratio (as a negative indicator for the freshness of POC) on surface methanogenesis, two PCAs were conducted with a) the focus on the upper 0-5 cmbsf, which is directly influenced by freshly sedimented organic material from the water column (Fig. 9), and b) the focus on the depth profiles throughout the sediment cores (up to 30 cmbsf) (Fig. 10).

For the upper 0-5 cmbsf in the sediment, a strong positive correlation was found between surface methanogenesis (integrated) and POC content (averaged) (Fig. 9c), indicating that POC content is an important controlling factor for methanogenesis in this layer. In support, highest bottom-near water chlorophyll concentrations coincided with highest bottom-near water methane concentrations and high integrated surface methanogenesis (0-5 cmbsf) in September 2013, probably as a result of the sedimentation of the summer phytoplankton bloom (Fig. 8). Indeed, the PCA revealed a strong positive correlation between integrated surface methanogenesis rates and bottom-near water methane concentrations (Fig. 9b) viewed over all investigated months. However, no correlation was found between bottom water chlorophyll and integrated surface methanogenesis rates (Fig. 9). As seen in Fig. 8, bottom-near high chlorophyll concentrations did not coincide with high bottom-near methane concentration in June/September 2014. We explain this result by a time lag between primary production in the water column and the export of the produced organic material to the seafloor, which was probably even more delayed during stratification. Such a delay was observed in a previous study (Bange et al., 2010), revealing enhanced water methane concentration close to the seafloor approximately one month after the chlorophyll maximum. The C/N ratio (averaged over 0-5 cmbsf) showed a weak positive correlation with integrated surface methanogenesis (0-5 cmbsf), which is surprising as we expected that a higher C/N ratio, indicative for less labile organic carbon, should have a negative effect on non-competitive methanogenesis. However, methanogens are not able to directly use most of the labile organic matter due their inability to process large molecules



(more than two C-C bondings) (Zinder, 1993). Methanogens are dependent on other microbial
groups to degrade large organic compounds (e.g. amino acids) for them (Zinder, 1993). Because of
this substrate speciation and dependence, a delay between the sedimentation of fresh, labile organic
matter and the increase in methanogenesis can be expected, which would not be captured by the
applied PCA.
In the PCA for the surface sediment profiles (0-30 cmbsf), POC showed a negative correlation with
methanogenesis, and sediment depth and C/N ratio showed a weak positive correlation with
methanogenesis (Fig 10.), which was also seen previously in the weak positive correlation between
integrated surface methanogenesis (0-5 cmbsf) and surface C/N (0-5 cmbsf). As POC, with the
exemption of the topmost sediment layer, remained basically unchanged over the top 30 cmbsf, its
negative correlation with methanogenesis is probably solely explained by the increase of
methanogenesis with sediment depth, and can therefore be excluded as a major controlling factor.
As sulfate in this zone was likely never depleted to levels that are critically limiting sulfate reduction
(lowest concentration 1300 μM, compare e.g. with Treude et al., 2014) we do not expect a significant
change in the competition between methanogens and sulfate reducers. It is therefore more likely
that the progressive degradation of organic matter into methanogenic substrates over depth and
time had a positive impact on methanogenesis. The C/N ratio indicates such a trend as the labile
fraction of POC decreased with depth. The mobilization of dissolved methanogenic substrates, such
as methanol, from organic matter would not be detectable by the C/N ratio as it is determined from
particulate samples.

### 4.3 Relevance of surface methanogenesis in Eckernförde Bay sediments

The time series station Boknis Eck in Eckernförde Bay is known for being a methane source to the
atmosphere throughout the year due to supersaturated waters, which result from significant benthic
methanogenesis and emission (Bange et al., 2010). The benthic methane formation is thought to take
place mainly in the deeper, sulfate-depleted sediment layers (Treude et al., 2005a; Whiticar, 2002).
In the present study, we show that surface methanogenesis within the sulfate zone is present despite
sulfate concentrations > 1 mM, a limit above which methanogenesis has been thought to be
negligible (Alperin et al., 1994; Hoehler et al., 1994; Burdige, 2006), and thus could contribute to
benthic methane emissions. In support of this hypothesis, high dissolved methane concentration in
the water column occurred with concomitant high surface methanogenesis activity (Fig. 8).
In fact, surface methanogenesis in the Eckernförde Bay could even increase in the future, as
temperature and oxygen, two important controlling factors identified for surface methanogenesis
(Maltby et al., 2016) and this study), are predicted to increase and decrease, respectively (Lennartz et
al., 2014), We will therefore have a closer look at the magnitude and potential relevance of this
process for methane the benthic methane budget.





Surface methanogenesis rates determined in the present study are in a similar range of other sulfate-
containing, organic-rich surface sediments (e.g. salt marsh sediments, sediments from the upwelling
region off Chile and Peru, or coastal sediments from Limfjorden, North Sea), (Table 2, References
herein). In comparison with methanogenesis rates below the sulfate-methane- transition zone
(SMTZ) of organic-rich sediments (coastal and upwelling sediments), rates were mainly lower (2-5
times) (Table 2), which is explained by the competition relief below the SMTZ, which makes more
substrates available for methanogenesis.
We also performed a comparison between surface (0-30 cmbsf) and deep (below the SMTZ) net
methanogenesis for the present study site to investigate the relevance of surface methanogenesis in
Eckernförde Bay sediments for the overall benthic methane budget.  In the gravity core of September
2013, the SMTZ was situated between 45 and 76 cmbsf (Fig. 3). The methane flux was estimated
according to Iversen & Jørgensen, (1993) using a sediment methane diffusion coefficient of $D_s =$
$1.64 \times 10^{-5}$ cm$^{-2}$ s$^{-1}$.  The sediment diffusion coefficient was derived from the seawater methane-
diffusion coefficient at 10 °C (Schulz, 2006), which was corrected by porosity according to Iversen &
Jørgensen, (1993). The calculated deep methane production (1.55 mmol m$^{-2}$ d$^{-1}$) was similar to earlier
calculated deep methanogenesis in Eckernförde Bay (0.66 – 1.88 mmol m$^{-2}$ d$^{-1}$; Treude et al., 2005a).
However, integrated hydrogenotrophic methanogenesis measured in the presented study below 45
cmbsf (determined by interpolation, 0.5 ± 0.2 mmol m$^{-2}$ d$^{-1}$) was up to 3 times lower compared to the
calculated deep methanogenesis, indicating that the interpolation missed hot spots of
hydrogenotrophic methanogenesis, as alternative pathways are not predicted for this zone given the
isotopic signature of methane (Whiticar, 2002). Surface methanogenesis in September 2013
represented 3-8 % of deep methanogenesis. While this percentage seems low, absolute surface
methanogenesis rates in Eckernförde Bay sediments are in the same magnitude as deep methane
production in other organic-rich sediments from the North Sea (0.076 mmol m$^{-2}$ d$^{-1}$, Jørgensen &
Parkes, 2010), or from the upwelling region off Chile (0.068-0.13 mmol m$^{-2}$ d$^{-1}$,Treude et al., 2005b),
indicating the general importance of this process. Compared to these other sites, Eckernförde Bay
features extremely high methanogenesis activity below the SMTZ, resulting in gas bubble formation
and ebullition (Abegg & Anderson, 1997; Jackson et al., 1998; Treude et al., 2005a).
How much of methane produced in the surface sediment is emitted into the water column depends
on the rate of methane consumption, i.e., aerobic and anaerobic oxidation of methane in the
sediment (Knittel & Boetius, 2009). In organic-rich sediments such as in the presented study, the oxic
sediment layer is often only mm-thick, due to the high rates of microbial organic matter degradation,
which rapidly consumes oxygen (Revsbech et al., 1980; Emerson et al., 1985; Jørgensen, 2006). Thus
the anaerobic oxidation of methane (AOM) might play a more dominant role in the present study. In
an earlier study from Eckernförde Bay, AOM rates were measured above the SMTZ (0-25 cmbsf), but



the authors concluded that it was fueled by deep methanogenesis (Treude et al., 2005a), as surface
integrated AOM rates (0.8-1.5 mmol m$^{-2}$ d$^{-1}$) were in the same magnitude as deep methane flux
(0.66-1.88 mmol m$^{-2}$ d$^{-1}$) from below the SMTZ (Treude et al., 2005a).
With the data set presented here we postulate that surface AOM above the SMTZ (0.8 mmol m$^{-2}$ d$^{-1}$,
Treude et al., (2005a) is mainly fueled by surface methanogenesis. If this is the case, then surface
methanogenesis is more likely in the range of 0.9 mmol m$^{-2}$ d$^{-1}$ (AOM + net surface methanogenesis),
indicating that surface methanogenesis could play a much bigger role for benthic methane budgeting
than previously thought. Whether surface methanogenesis at Eckernförde Bay has the potential for
direct methane emissions into the water column goes beyond the informative nature of our dataset
and should be tested in future studies. Our study shows that surface methanogenesis correlates with
methane concentrations in the water column near the seafloor; however, so could also
methanogenesis and gas ebullition from below the SMTZ.
## 5. Summary
The present study demonstrated that methanogenesis and sulfate reduction were concurrently
active within the sulfate-reducing zone in sediments at Boknis Eck (Eckernförde Bay, SW Baltic Sea).
Observed methanogenesis was probably based on non-competitive substrates due to the
competition with sulfate reducers for the substrates H$_2$ and acetate. Accordingly, members of the
family *Methanosarcinaceae*, which are known for methylotrophic methanogenesis and were found in
the surface sediments, are likely to be responsible for the observed surface methanogenesis using
the substrates methanol, methylamines or methylated sulfides.
An important factor controlling surface methanogenesis in the upper 0-5 cmbsf was the POC content,
resulting in highest methanogenesis activity after summer and autumn phytoplankton blooms.
Increased stratification (indicated by increased salinity at the seafloor) was also found to be
beneficial for surface methanogenesis, as it leads the decline of oxygen below the pycnocline.
Accordingly, oxygen depletion during later summer showed a strong positive correlation with surface
methanogenesis, enabling more organic matter to reach the seafloor and providing a larger habitable
anoxic zone for methanogens in the surface sediment.
With increasing sediment depth (0-30 cmbsf), methanogenesis revealed only a positive correlation
with C/N ratio, indicating that a progressive mobilization of dissolved methanogenic substrates from
fermentation plays an important role for controlling non-competitive methanogenesis.
Even though surface methanogenesis was low compared to methanogenesis below the STTZ, it may
play an underestimated role in the methane budget at Boknis Eck, e.g., by directly fueling AOM
above the SMTZ.





**Author Contribution**
J.M. and T.T. designed the experiments. J.M. carried out all experiments. H.W. coordinated
measurements of water column methane and chlorophyll. C.L. and M.F. conducted molecular
analysis. M.S. coordinated 13C-Isotope measurements. J.M. prepared the manuscript with
contributions from all co-authors.
**Data Availability**
Research data for the present study can be accessed via the public data repository PANGEA
(doi:10.1594/PANGAEA.873185).
**Acknowledgements**
We thank the captain and crew of F.S. Alkor, F.K. Littorina and F.B. Polarfuchs for field assistance. We
thank G. Schüssler, F. Wulff, P. Wefers, A. Petersen, M. Lange, and F. Evers for field and laboratory
assistance. For the geochemical analysis we want to thank B. Domeyer, A. Bleyer, U. Lomnitz, R.
Suhrberg, and V. Thoenissen. We thank F. Malien, X. Ma, A. Kock and T. Baustian for the $O_2$, $CH_4$, and
chlorophyll measurements from the regular monthly Boknis Eck sampling cruises. Further we thank
R. Conrad and P. Claus at the MPI Marburg for the $^{13}$C-Methanol measurements. This study received
financial support through the Cluster of Excellence "The Future Ocean" funded by the German
Research Foundation, through the Sonderforschungsbereich (SFB) 754, and through a D-A-CH project
funded by the Swiss National Science Foundation and German Research foundation (grant no.
200021L_138057, 200020_159878/1). Further support was provided through the EU COST Action
PERGAMON (ESSEM 0902), through the BMBF project BioPara (grant no. 03SF0421B) and through
the EU's H2020 program (Marie Curie grant NITROX # 704272 to CRL).

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



**Figure Captions**

**Figure 1:** Parameters measured in the water column and sediment at each sampling month in the year 2013. Net methanogenesis (MG) and hydrogenotrophic (hydr.) methanogenesis rates are shown in triplicates with mean (solid line).

**Figure 2**: Parameters measured in the water column and sediment at each sampling month in the year 2014. Net methanogenesis (MG) and hydrogenotrophic (hydr.) methanogenesis rates are shown in triplicates with mean (solid line).

**Figure 3:** Parameters measured in the sediment in the gravity core in September 2013. Hydrogenotrophic (hydr.) methanogenesis rates are shown in triplicates with mean (solid line).

**Figure 4:** Integrated net methanogenesis (MG) rates and hydrogenotrophic MG rates (0-25 cmbsf) for each time point.

**Figure 5:** Potential methanogenesis rates of the four different treatments in November 2013, March 2014, June 2014 and September 2014. Control (blue symbols) is describing the treatment with sediment plus artificial seawater containing natural salinity (24 PSU) and sulfate concentrations (17 mM), molybdate (green symbols) is the treatment with addition of molybdate (22 mM), BES (purple symbols) is the treatment with 60 mM BES addition, and methanol (red symbols) is the treatment with addition of 10 mM methanol. Shown are triplicates per depth interval and the mean as a solid line. Please note the different x-axis for the methanol treatment (red).

**Figure 6**: Concentrations (A) and isotope composition (B) of porewater methanol ($CH_3OH$), headspace methane ($CH_4$), and headspace carbon dioxide ($CO_2$) during the sediment-slurry experiment (with sediment from the 0-1 cmbsf horizon in September 2014) with addition of $^{13}C$-enriched methanol ($^{13}C$:$^{12}C$ = 1:1000). Experiment was conducted over 37 days at in-situ temperature (13°C). Shown are means (from triplicates) with standard deviation.

**Figure 7:** Sediment methane concentrations over time in the treatment with addition of methanol (10 mM) are shown above. Shown are triplicate values per measurement. DNA copies of *Archaea*, *Methanosarcinales* and *Methanosarcinaceae* are shown below in duplicates per measurement. Please note the secondary y-axis for *Methanosarcinales* and *Methanosarcinaceae.* More data are available for methane (determined in the gas headspace) than from DNA samples (taken from the sediment) as sample volume for molecular analyzes was limited.

**Figure 8:** Temporal development of integrated net surface methanogenesis (0-5 cmbsf) in the sediment and chlorophyll (green) and methane concentrations (orange) in the bottom water (25 m).





Methanogenesis (MG) rates and methane concentrations are shown in means (from triplicates) with
standard deviation.
**Figure 9:** Principle component analysis (PCA) from three different angles of integrated surface
methanogenesis (0-5 cmbsf) and surface particulate organic carbon averaged over 0-5 cmbsf (surface
sediment POC), surface C/N ratio averaged over 0-5 cmbsf (surface sediment C/N), bottom water
salinity, bottom water temperature (T), bottom water methane ($CH_4$), bottom water oxygen ($O_2$), and
bottom water chlorophyll. Data were transformed into ranks before analysis. a) Correlation biplot of
principle components 1 and 2, b) correlation biplot of principle components 1 and 3, c) correlation
biplot of principle components 2 and 3. Correlation biplots are shown in a multidimensional space
with parameters shown as green lines and samples shown as black dots. Parameters pointing into
the same direction are positively related; parameters pointing in the opposite direction are
negatively related.

**Figure 10:** Principle component analysis (PCA) from two different angles of surface methanogenesis
depth profiles and sampling month (Month), sediment depth, depth profiles of particulate organic
carbon (POC) and C/N ratio (C/N). Data was transformed into ranks before analysis. a) Correlation
biplot of principle components 1 and 2, b) correlation biplot of principle components 1 and 3.
Correlation biplots are shown in a multidimensional space with parameters shown as green lines and
samples shown as black dots. Parameters pointing into the same direction are positively related;
parameters pointing in the opposite direction are negatively related.










**Table 1:** Sampling months with bottom water (~ 2 m above seafloor) temperature (Temp.), dissolved
oxygen ($O_2$) and dissolved methane ($CH_4$) concentration

| Sampling Month | Date | Instrument | Temp. (°C) | $O_2$ (µM) | $CH_4$ (nM) | Type of Analysis |
|---|---|---|---|---|---|---|
| March 2013 | 13.03.2013 | CTD | 3 | 340 | 30 | WC |
|  |  | MUC |  |  |  | All |
| Juni 2013 | 27.06.2013 | CTD | 6 | 94 | 125 | WC |
|  |  | MUC |  |  |  | All |
| September 2013 | 25.09.2013 | CTD | 10 | bdl | 262* | WC |
|  |  | MUC |  |  |  | All |
|  |  | GC |  |  |  | GC-All |
| November 2013 | 08.11.2013 | CTD | 12 | 163 | 13 | WC |
|  |  | MUC |  |  |  | All |
| March 2014 | 13.03.2014 | CTD | 4 | 209 | 41* | WC |
|  |  | MUC |  |  |  | All |
| June 2014 | 08.06.2014 | CTD | 7 | 47 | 61 | WC |
|  |  | MUC |  |  |  | All |
| September 2014 | 17.09.2014 | CTD | 13 | bdl | 234 | WC |
|  |  | MUC |  |  |  | All |

MUC = multicorer, GC= gravity corer, CTD = CTD/Rosette, bdl= below detection limit (5µM), All = methane gas
analysis, porewater analysis, sediment geochemistry, net methanogenesis analysis, hydrogenotrophic
methanogenesis analysis, GC-All= analysis for gravity cores including methane gas analysis, porewater analysis,
sediment geochemistry, hydrogenotrophic methanogenesis analysis, WC= Water column analyses including
methane analysis, chlorophyll analysis
**Concentrations from the regular monthly Boknis Eck sampling cruises on 24.09.13 and 05.03. 14 (www.bokniseck.de)











**Table 2:** Comparison of surface methanogenesis rates in shallow water marine sediments of different
geographical origin

| Study site | Water depth (m) | Sediment depths (cm) | Rate (nmol cm$^{-3}$ d$^{-1}$) | Reference |
|---|---|---|---|---|
| *Sulfate-containing, organic-rich sediments* | | | | |
| Eckernförde Bay (Baltic Sea) | 28 | 0-25 | 0 -1.3 | Present study |
| Upwelling region off Peru (Pacific) | 70-1025 | 0-25 | 0-1.5 | (Maltby et al., 2016) |
| Upwelling region off Chile (Pacific) | 87 | 0-6 | 0-0.6 | (Ferdelman et al., 1997) |
| Limfjorden (North Sea) | 7-10 | 0-100 | 0-0.05 | (Jørgensen & Parkes, 2010) |
| Colne Point Saltmarsh (Essex, UK) | - | 0-30 | 0-0.03 | (Senior et al., 1982) |
| *Sulfate-depleted, organic-rich sediments (sediment depth marks the depth at which sulfate was depleted)* | | | | |
| Eckernförde Bay (Baltic Sea) | 28 | > 100 | 0.01-1.4 | Present Study |
| Limfjorden (North Sea) | 7-10 | > 100 | 0.01-3.1 | (Jørgensen & Parkes, 2010) |
| Saanich Inlet (British Columbia, Canada) | 225 | > 20 | 0.3-7.0 | (Kuivila et al., 1990) |
| Upwelling region off Peru (Pacific) | 78 | > 50 | 0-2.1 | (Maltby et al., 2016) |













Figures
**Figure 1**

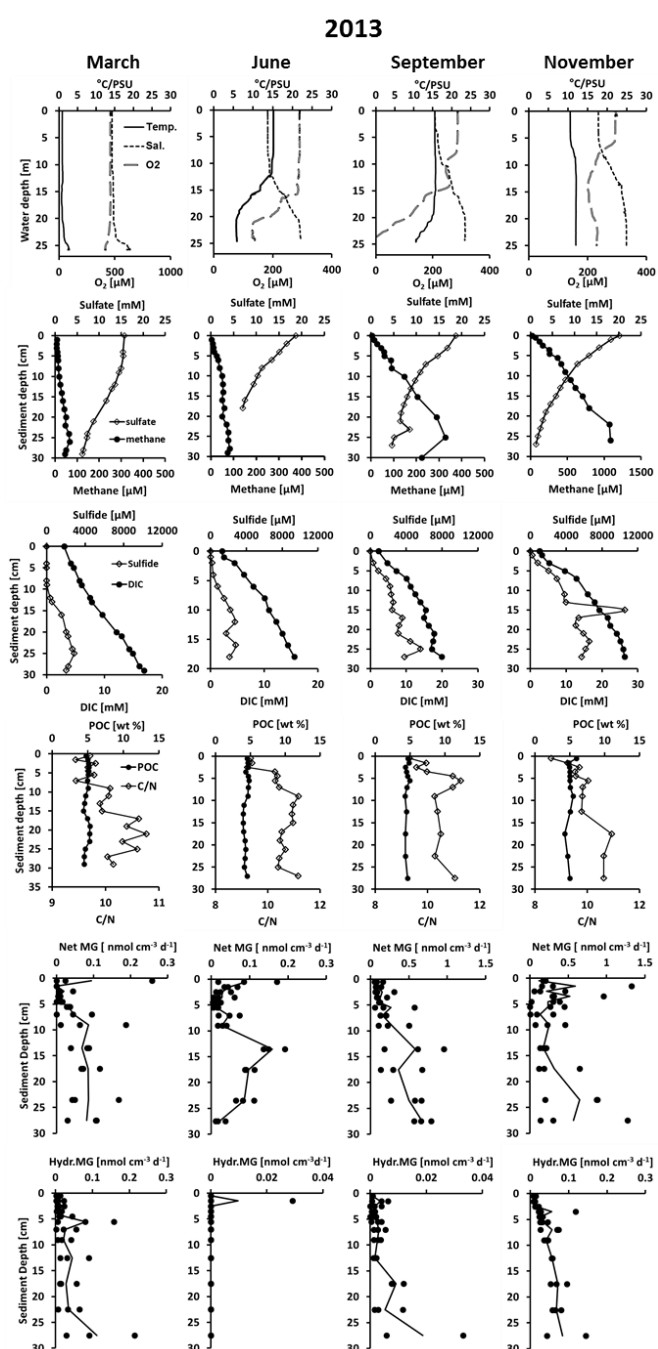





**Figure 2**

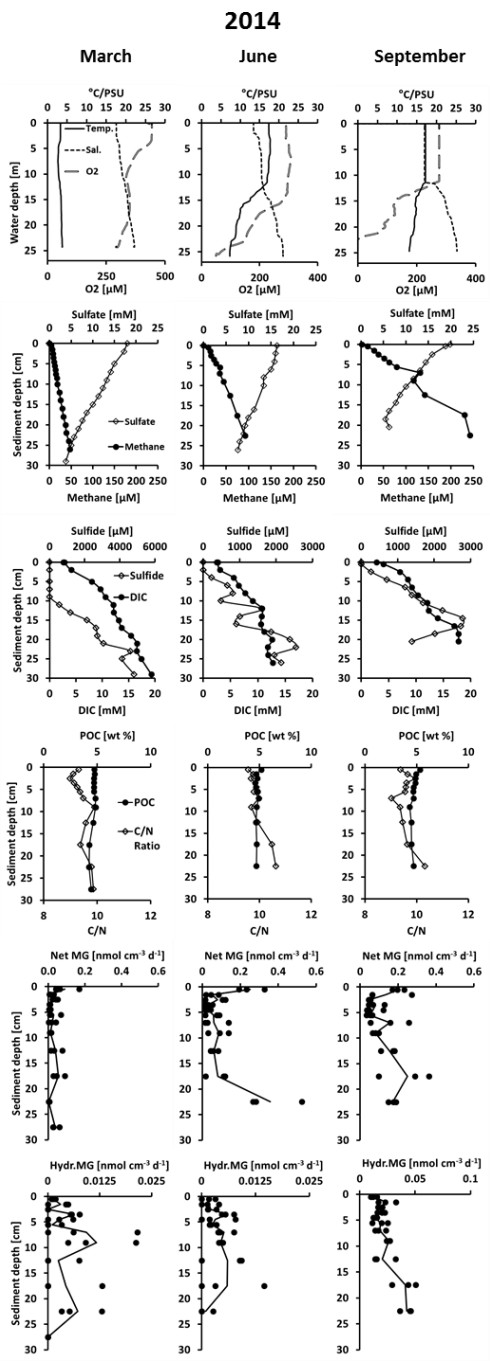







**Figure 3**


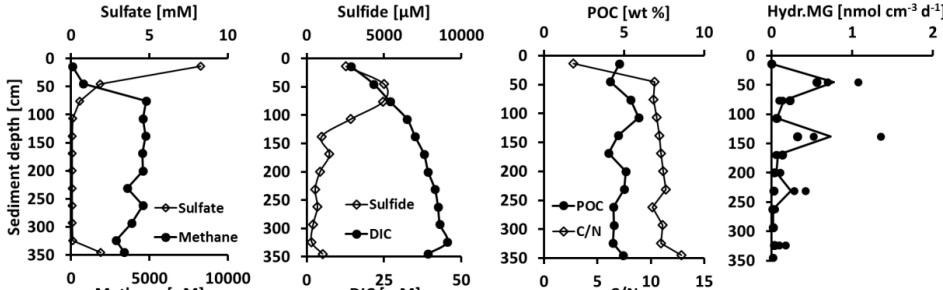






















**Figure 4**

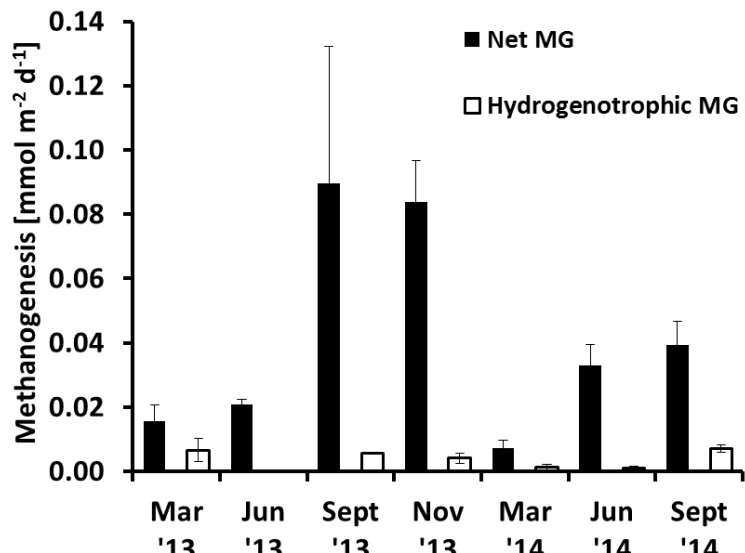















**Figure 5**

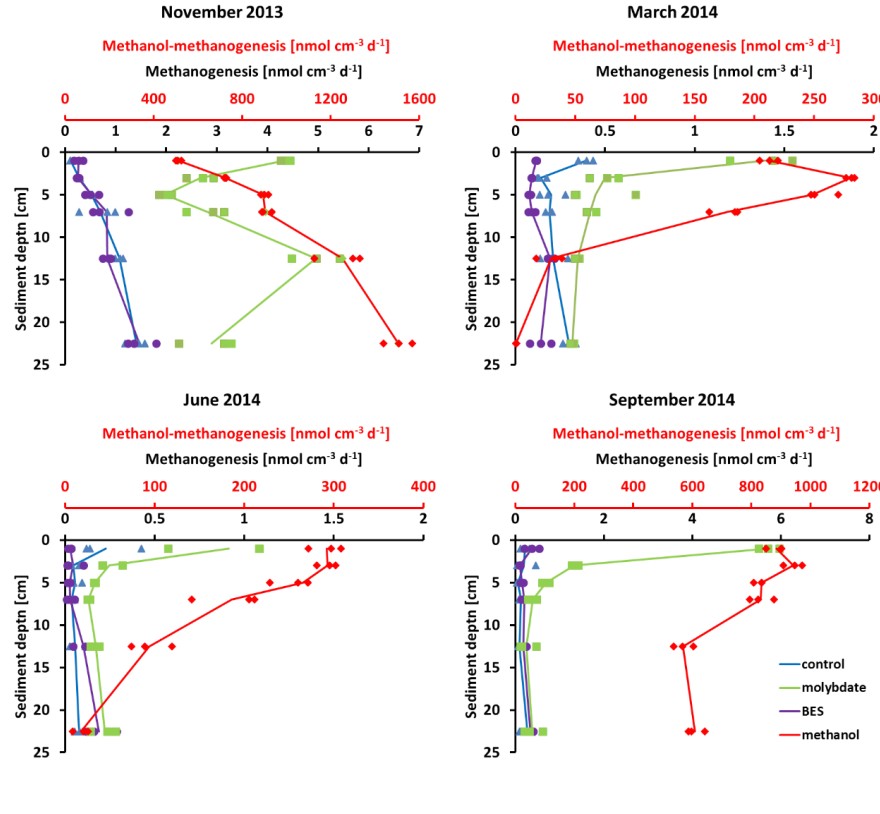














**Figure 6**

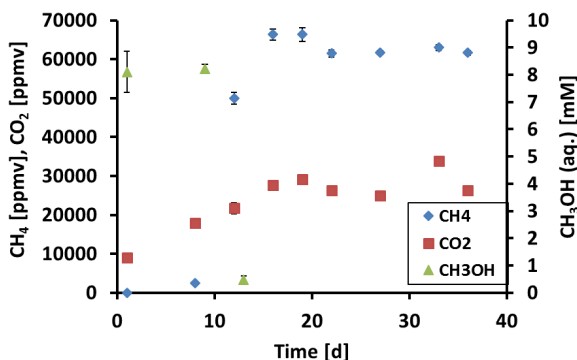

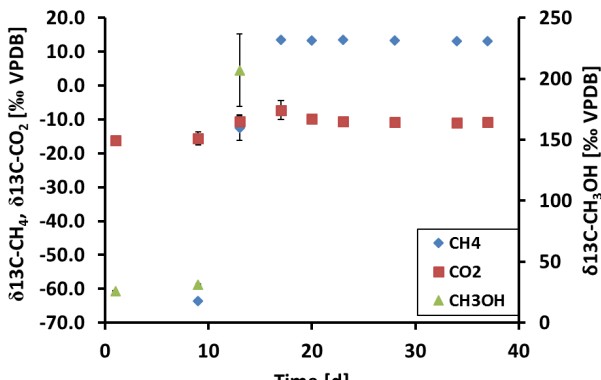















**Figure 7**

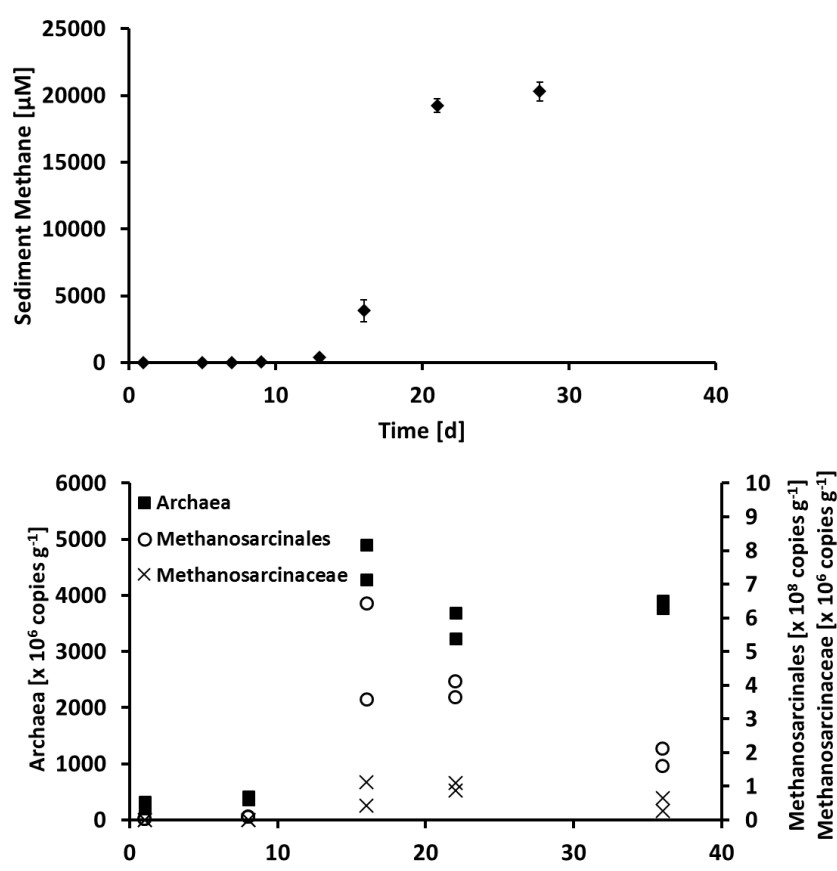














**Figure 8**

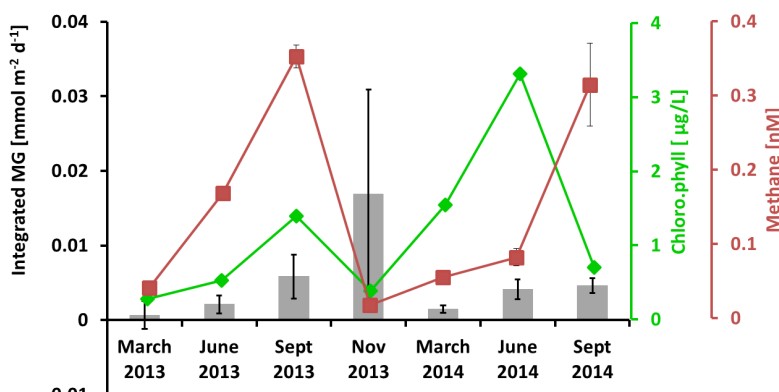




















**Figure 9**



**Figure 10**

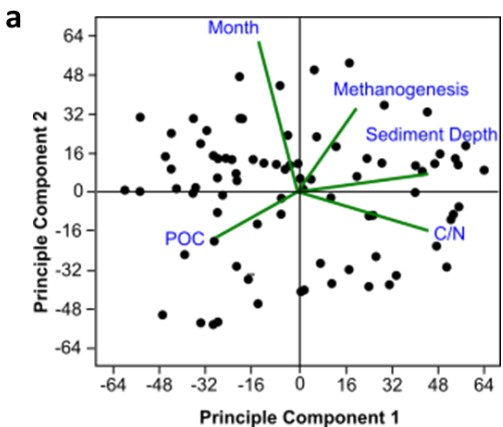

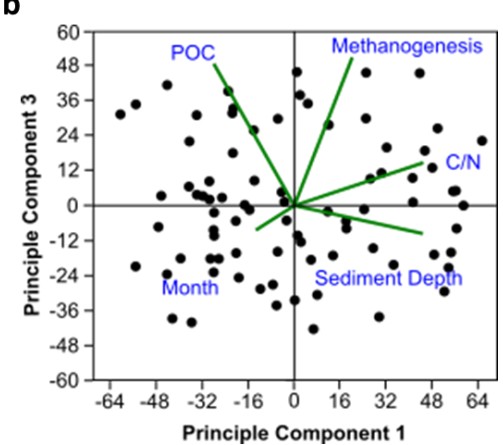







