# Peer review of "Microbial methanogenesis in the sulfate-reducing zone of sediments in"

_Biogeosciences, 2017_

## Referee Comment (RC1) · Anonymous Referee #1 · 16 May 2017

Shallow littoral sediments are a poorly constrained source of methane to marine and brackish water columns. Normally, methane fluxes from marine sediments into the water column are restricted by the large fluxes of sulfate available to microbial sulfate reduction taking place in the sediments. This "microbial lid" on methane effluxes derives in part from the competitive advantage of organoclastic sulfate reducing bacteria versus methanogens for buried reactive organic carbon substrates, and also to the direct oxidation of upward diffusing methane by methanotrophic sulfate reducing prokaryotes. However, methane ebullition from deeper layers into the surface sediments, or the production of methane from non-competitive substrates (e.g. methyl amines, or methanol) may contribute significantly to the methane flux into bottom waters. It is the latter pro-

cess that the authors of this study seek to address and quantify in Eckernförde Bay sediments. Their approach can be divided into two parts: 1) a seasonal study of sediment methane biogeochemistry, including rate measurements, and 2) an experimental enrichment to examine the effect of methanol as a potential non-competitive substrate in Bognis Eck sediments.

Maltby and co-authors present a detailed seasonal data set showing geochemical and experimental data collected over two years from the shallow, organic-rich sediments of Eckernförde Bay in the Baltic Sea. Although it has been known now for decades that minor amounts of methane forms in sulfate-reducing sediments from methanogenesis of non-competitive substrates, the role that this process plays in Eckernförde deep waters was not clear prior to this study. The data and outcome of the present study are consistent with previous studies of methanogenesis using non-competitive substrates and suggests that methane derived from non-competitive substrates may be a source of methane for the Eckernförde deep water. This study adds to the data and knowledge concerning sediment biogeochemical processes for Bognis Eck, which has been the site of a successful string of studies investigating the biogeochemistry of deep anoxic waters and the underlying sediments in Eckernförde Bay. The geochemical data is of high quality. The down core experimental tracer data is also of good quality, although I have reservations about interpretation of some of the experiments (see Major Issues below).

Nevertheless, there are a number of points in the manuscript that the authors need to address.

Major issues:

1. Section 3.4.1 (and Methods – lines 235-244) Net methanogenesis: These rates do not necessarily represent methanogenesis in the presence of sulfate. Were the sulfate concentrations monitored during the incubations? There are no time course data of sulfate (nor methane) shown for these experiments. As the incubations were

performed over four weeks, the chances that sulfate became depleted within several days at many of the depths is very likely given SR rates of up to 10000 nmol cmˆ3 dayˆ-1. Therefore, the direct comparison of the 14C labeled hydrogenotrophic rates with Net Methanogenesis rates are not at all valid. The Net MG rates are very likely a severe overestimation of actual in situ rates of methanogenesis.

Likewise, the Manipulated Methanogenesis experiments are not described in enough detail to evaluate them properly. Were these experiments performed like the Net Methanogenesis experiments? Or were they performed over shorter period of time using radiolabeled bicarbonate?

2. I am not sure how insightful the 13C-labeled methanol enrichments are for understanding the role of non-competitive substrates at this site. First of all, no in situ methanol concentrations are provided. Secondly, and more importantly, the authors added methanol up to 10 mM. These are enrichment concentrations that are not likely to reflect environmental conditions. Enrichment, or growthn methanol, is what they see in the experiments, as shown in Figures 6 and 7. The conclusion that these enriched organisms represent the in situ organisms and metabolisms is not tenable. This experiment does not even shed light on whether or not there was non-competitive methanogenesis occurring in the experiments slurries themselves. What happened to sulfate during this experiment? Was there still sulfate present after 10 days?

3. The Discussion needs to be made more concise. The authors should directly address the stated main point of the manuscript: Is there methanogenesis in the sulfate reducing zone, does it proceed via non-competitive substrates, and is it at all important for methane fluxes to the deep water? The discussion as written now is, to a large extent, a reiteration of the results with some commentary. It also tends to drift off into unwarranted speculation. Some parts that could be excised without detriment:

a. Lines 564 and following : "possible" additional sources of carbon and the production of hydrogen

b. Lines 626 "Reaction of sulfide with methyl groups and organic matter. . .discussion is beside the point.

c. Lines 646 Discussion of dissolution of CO2 in water was already discussed earlier in Results.

d. Section 670 The discussion on temperature is speculative and I am not sure where it is leading.

e. Lines 783 and following: The discussion of deep methanogenesis (below the SMTZ) appears to be beyond the scope of the manuscript (i.e. methanogenesis in surface sediments)

One means of shortening the discussion might be to delete or severely scale-back to the discussion revolving around the PCA analysis. I do not see how the analysis and resulting discussion adds anything new to our understanding of the controls on methanogenesis in marine sediments. In considering such a discussion, it might be worth for the authors to revisit the seminal articles on this topic by Crill and Martens (L&O 1983 and GCA 1986).

Specific Comments:

Line 282. This sentence is confusing. "Fast oxygen consumption" does not correlate with "slowed microbial activity".

Figures 1 and 2. The postage stamp size plots (at least in the BG Discussions version) are difficult to read. Perhaps taking he water column data out and combining it into a separate figure would help?

Lines 424-434. I would not put so much emphasis on the single bottom points of the gravity core.

Line 469: The hydrogenotrophic methanogenic activity at 45 cm depth at the sulfate-methane transition zone may be in part due to tracer back flux associated with AOM

(see Holler et al., PNAS 2011).

Figure 7: What is the difference between the methane concentration in this figure and in Figure 6? Why not combine Figures 6 & 7?

Lines 525 and following: What are the criteria for calling something a "strong" or "weak" correlation. Line 554 It might be good to briefly describe how BES works as an inhibitor, and why it has no effect here.

Line 566 How deep is bioturbation in Bognis Eck? And was the shell at 20 cm living or just debris? Figure 8: Based on what criteria was 0-5 cm depth for integrated methanogenesis chosen, whereas, similar data, but from 0-25 cm is shown in Figure 4?

Line 614: Again, this looks like a growth curve.

Line 637: These organisms became dominant due to the highly enriched methanol concentrations employed. This does not say anything about their importance under in situ conditions.

Line 690 and following: Changing sulfate concentration-depth profiles as a response to changing salinity conditions indicates that this is a non-steady-state situation. Ergo, it is not possible to use this as an indication of microbial sulfate reduction.

Line 841: How does the fueling of AOM above the SMTZ cause methanogenesis to play an "underestimated" role? I would expect that AOM would minimize the impact of methanogenesis on the water column methane budget.

Technical comments:

Line 138 "that" instead of "which"

Line 437 "Content" not "concentration" for POC wt%

Line 612: Sentence is confusing: "of" rather than "if"? Also, the population changes

to the new conditions; you do not have any evidence for adaption (and evolutionary concept).

---

## Referee Comment (RC2) · Anonymous Referee #2 · 15 Jun 2017

The work presented by Maltby et al. is really nice piece of study gathering results from several impressive campaign of sampling and involving different cutting-edge methods. Their findings give an interesting overview of biological processes and environmental factors controlling methane emissions from sediments and water column of a Baltic sea bay, well known for its importance in global methane emissions. The originality of their work lies in the demonstration of co-occurence of sulphate reduction and methanogenesis in surface sediments. This co-existence is permitted by a mechanism developed by some methanogenesis microorganisms to escape from the strong competition with sulphate-reducing microorganisms: using (releasing? I did not find information on that) non-competitive substrates. The manuscript is overall well written except the abstract,

see my comments below. I have only two main concerns.

First, the article is sometime written in a way that only initiates of the field may touch. The first sentence of abstract directly starts with the work done without putting the study in a wider context. The object you study is complex and well structured. We do not immediately understand the relevance of studying methanogenesis in the sulphate-reducing zone. We neither understand that you studied surface, deep sediment and the water column and not only surface (sulphate-reducing) sediment. The reference of "a non-competitive substrate" is not understandable. Which competition do you refer ? Implying which organisms? In the introduction, it could be useful to build a synthetic figure summarizing the studied ecosystem including the different compartments, different organisms, interactions among organisms, exchanges of matters between these compartments.

Second, the (minor) contribution of surface methanogenesis to total methane emissions from this ecosystem is a bit hidden in the article. This contribution deserves to be clearly presented in the abstract. To my point of view, the minor contribution of this mechanism does not question the quality and relevance of this study, and is an important information. In the same vein, the statement that surface methanogenesis could play a key role in fueling the surface anaerobic oxidation of methane is speculative since this last process was not measured in the study.

Specific comments.

Line 30 supress "in the manipulated experiments. L31-33 this new objective that pops up too late. Please gather your objective in one sentence L47 replace "makes an important contribute" by "substantially contributes to". I did not understand the last part "as it could..." L78-79 and throughout the manuscript. The expression "Environmental control mechanisms" is a bit elusive. Do you mean "environmental controls" or "biological processes"? Try to better specify what should be better studied. L164 Rewrite your sentence to clarify. Could be "Biological activities of samples were stopped by

the addition of mercury chloride solution..." L177 "extracted"? you mean "sampled" for analysis? L193-216 I am not expert in measurement of methane concentration in sediment, but I am wondering whether the fact of cutting sediment core in 1 cm sediment interval could release, at least a part of, the methane you wish to quantify. L236 Could you rapidly explain again what is the hydrogenotrophic methanogenesis? And what is the interest of measuring this in the context of your study? L389 Supress "in" before september. L401 I guess you're talking about the C/N of particulate organic matter, but I am not sure. Please specify. L543 I propose you to replace the end of your sentence by "...the following observations that will be discussed in more detail in the following chapters". My first reaction was to try to understand your arguments before reading the following chapters. L569 Your explanations about the competition between sulphate-reducing and methanogenesis microorganisms, and the strategy of methylotrophic methatogeners to escape from this competition, are very clear and convincing. Now I am wondering whether there is competition between hydrogenotrophic and methyltrophic microorganisms. And if yes, does this competition change with depth? L587-592 This sentence is too long. Split your explanations into 2 sentences. L614 If I follow well, you should add a "P" after "DMS". L605-631 Maybe this is a limit of your study of not having quantified some key non-competitive substrates in sediments and water. It could be discussed in a paragraph drawing next investigations that could be done. L640 What fractionation are you discussing? An isotopic fractionation? You must better explain. L488-489 Could you check whether such moderate isotopic fractionation (factor of 1.07-1.08) could explain an increase of delta of almost 200 per mille. I have a doubt. L644-645 This sentence is not clear. What would be the alternative explanation? L646 One bracket is lacking at the end of sentence. L684-694 I did not understand your explanations. Please try to reformulate and be more direct when you propose an interpretation. L706-707 Did you find results going in this way as well? L713-736 This section is really too long. Split it in two paragraphs, one focusing on the effect of POC amount and the other on C/N ratio. L830-831 I do not understand your interpretation of the positive correlation between surface methanogenesis and C/N ratio

of POM. L805-809 and 832-834. This process of anaerobic consumption of methane (AOM) was not measured in this study making all these discussions around the key role of surface methanogenesis in fuelling AOM very speculative. I do not understand why deep methanogenesis, which contributes for the major part of methane emissions, does not contribute to AOM fueling. It sounds like you would absolutely like to give a central importance to surface methanogenesis.

---

## Author Comment (AC1) · 14 Sep 2017

We would like to thank the reviewer for her/his critical comments, which we think helped to improve the quality and clarity of this manuscript. We hope our responses and adaptations are adequate to accept this manuscript for publication in Biogeosciences. Please find our detailed responses below.

Anonymous Referee #1

Shallow littoral sediments are a poorly constrained source of methane to marine and brackish water columns. Normally, methane fluxes from marine sediments into the water column are restricted by the large fluxes of sulfate available to microbial sulfate reduction taking place in the sediments. This "microbial lid" on methane effluxes derives in part from the competitive advantage of organoclastic sulfate reducing bacteria versus methanogens for buried reactive organic carbon substrates, and also to the direct oxidation of upward diffusing methane by methanotrophic sulfate reducing prokaryotes. However, methane ebullition from deeper layers into the surface sediments, or the production of methane from non-competitive substrates (e.g. methyl amines, or methanol) may contribute significantly to the methane flux into bottom waters. It is the latter pro- cess that the authors of this study seek to address and quantify in Eckernförde Bay sediments. Their approach can be divided into two parts: 1) a seasonal study of sediment methane biogeochemistry, including rate measurements, and 2) an experimental enrichment to examine the effect of methanol as a potential non-competitive substratein Bognis Eck sediments. Maltby and co-authors present a detailed seasonal data set showing geochemical and experimental data collected over two years from the shallow, organic-rich sediments of Eckernförde Bay in the Baltic Sea. Although it has been known now for decades that minor amounts of methane forms in sulfate-reducing sediments from methanogenesis of non-competitive substrates, the role that this process plays in Eckernförde deep waters was not clear prior to this study. The data and outcome of the present study are consistent with previous studies of methanogenesis using non-competitive substrates and suggests that methane derived from non-competitive substrates may be a source of methane for the Eckernförde deep water. This study adds to the data and knowledge concerning sediment biogeochemical processes for Bognis Eck, which has been the site of a successful string of studies investigating the biogeochemistry of deep anoxic waters and the underlying sediments in Eckernförde Bay. The geochemical data is of high quality. The down core experimental tracer data is also of good quality, although I have reservations about interpretation of some of the experiments (see Major Issues below). Nevertheless, there are a number of points in the manuscript that the authors need to address.

Major issues: 1. Section 3.4.1 (and Methods – lines 235-244) Net methanogenesis: These rates do not necessarily represent methanogenesis in the presence of sulfate. Were the sulfate concentrations monitored during the incubations? There are no time course data of sulfate (nor methane) shown for these experiments. As the incubations were performed over four weeks, the chances that sulfate became depleted within several days at many of the depths is very likely given SR rates of up to 10000 nmol cmËE̦3 dayËE̦-1. Therefore, the direct comparison of the 14C labeled hydrogenotrophic rates with Net Methanogenesis rates are not at all valid. The Net MG rates are very likely a severe overestimation of actual in situ rates of methanogenesis.

Authors Reply: Thank you for initiating this discussion. We will add all methane development graphs to the supplementary material. Measurements of sulfate during the incubation was unfortunately not possible as we worked with closed headspace systems. We agree that sulfate likely declined during the incubations as we expected simultaneous sulfate reduction activity and because no sulfate was supplied from the overlying water. However, if sulfate would have been completely depleted over the course of the incubation, we would have expected a change in the steepness of the methane production, i.e. an increase in methane production after sulfate was exhausted, which was not the case. Secondly, a quick calculation using sulfate reduction rates from a past study (Bertics et al. 2013) and our sulfate concentrations tells us that sulfate was unlikely depleted. In the surface sediment, where sulfate reduction is usually highest (∼200 nmol dm-3 d-1) together with sulfate (∼18 $\mu$mol cm-3), sulfate would have theoretically been depleted to 12 $\mu$mol cm-3 within a 30 day incubation. In the deepest layers, where sulfate reduction is lowest (∼10 nmol dm-3 d-1) together with sulfate (∼2 $\mu$mol cm-3), sulfate would have theoretically been depleted to 1.7 $\mu$mol cm-3 within a 30 day incubation. As these calculations illustrate, we do not expect total exhaustion of sulfate and we also see no evidences for this in the methane production rates.

2. Likewise, the Manipulated Methanogenesis experiments are not described in enough detail to evaluate them properly. Were these experiments performed like the

Net Methanogenesis experiments? Or were they performed over shorter period of time using radiolabeled bicarbonate?

Authors Reply: We agree with the reviewer that some critical information concerning the methods for the manipulated experiment (e.g. incubation time) were missing. These experiments were performed like the net methanogenesis rates (besides the manipulation) and we have added this information to the methods section.

The handling of samples and incubations for the determination of hydrogenotrophic methanogenesis is described in chapter 2.7.1.2 "Hydrogenotrophic Methanogenesis".The handling of samples with radiotracer was different from net methanogensis given the different nature of this rate determination method.

3. I am not sure how insightful the 13C-labeled methanol enrichments are for understanding the role of non-competitive substrates at this site. First of all, no in situ methanol concentrations are provided. Secondly, and more importantly, the authors added methanol up to 10 mM. These are enrichment concentrations that are not likely to reflect environmental conditions. Enrichment, or growthn methanol, is what they see in the experiments, as shown in Figures 6 and 7. The conclusion that these enriched organisms represent the in situ organisms and metabolisms is not tenable. This experiment does not even shed light on whether or not there was non-competitive methanogenesis occurring in the experiments slurries themselves. What happened to sulfate during this experiment? Was there still sulfate present after 10 days?

Authors Reply: Regarding sulfate concentration, please see our comment above. We are aware that methanol concentrations applied in our study were higher than usually observed under environmental conditions. The purpose of this experiment was to investigate the potential of the methanogenic community to use methanol as a non-competitive substrate. Other studies did similar stimulation experiments to demonstrate the activity of this metabolic group in the presence of sulfate reduction (see e.g. Oremland and Polcin 1982). We can of course not make any statement, how important this process is under in situ conditions in relation to those that use other non-competitive substrates for methanogenesis, which we now discuss in more detail.

4. The Discussion needs to be made more concise. The authors should directly address the stated main point of the manuscript: Is there methanogenesis in the sulfate reducing zone, does it proceed via non-competitive substrates, and is it at all important for methane fluxes to the deep water? The discussion as written now is, to a large extent, a reiteration of the results with some commentary. It also tends to drift off into unwarranted speculation. Some parts that could be excised without detriment:

a. Lines 564 and following : "possible" additional sources of carbon and the production of hydrogen Authors Reply: We believe that this discussion is important to explain the observed higher rates in March; hence, we would like to keep it.

b. Lines 626 "Reaction of sulfide with methyl groups and organic matter. . .discussion is beside the point. Authors Reply: We agree that this part is too extensive and deleted it.

c. Lines 646 Discussion of dissolution of CO2 in water was already discussed earlier in Results. Authors Reply: We deleted this part as it is repetition.

d. Section 670 The discussion on temperature is speculative and I am not sure where it is leading. Authors Reply: We do think that the positive correlation between temperature and methanogenesis is an important point to mention, as temperature is a strong environmental factor in this temperate environment, which could explain some of the variations in methanogenesis. We clarified this.

e. Lines 783 and following: The discussion of deep methanogenesis (below the SMTZ) appears to be beyond the scope of the manuscript (i.e. methanogenesis in surface sediments)

Authors Reply: As the majority of methanogenesis occurs below the SMTZ, we believe it is very important to compare both surface and deep methanogenesis and their potential to emit methane into the water column for assessing the relevance of surface methanogenesis.

One means of shortening the discussion might be to delete or severely scale-back to the discussion revolving around the PCA analysis. I do not see how the analysis and resulting discussion adds anything new to our understanding of the controls on methanogenesis in marine sediments. In considering such a discussion, it might be worth for the authors to revisit the seminal articles on this topic by Crill and Martens (L&O 1983 and GCA 1986).

Authors Reply: In our opinion, the PCA analysis is crucial, as it gives us statistical security about the potential environmental controls on surface methanogenesis. Without a statistical analysis, the discussion about environmental controls would be very speculative. Our study brings new insights into environmental controls, as we are one of the first ones studying environmental controls on surface methanogenesis within the sulfate-reducing zone. Therefore, we like to keep this part.

Specific Comments: Line 282. This sentence is confusing. "Fast oxygen consumption" does not correlate with "slowed microbial activity".

Authors Reply: What we meant was that due to quick exhaustion of oxygen in the core after retrieval, i.e. after capping the core from oxygen supply, organic matter degradation shifted to slower anaerobic processes. We clarified this in the manuscript.

Figures 1 and 2. The postage stamp size plots (at least in the BG Discussions version) are difficult to read. Perhaps taking he water column data out and combining it into a separate figure would help?

Authors Reply: Separating the water column data from the other profiles would make it harder for the reader to the connection between e.g. possible oxygen depletion in the water column and high surface methanogenesis rates. We therefore would like to keep it together. To make it easier to read, we increased the font size on the plots and also left out some redundant axis titles.

Lines 424-434. I would not put so much emphasis on the single bottom points of the gravity core.

Authors Reply: We think it is necessary to discuss the increase in sulfate at 350 cmbsf , as sulfate is a crucial factor for methanogenesis..

Line 469: The hydrogenotrophic methanogenic activity at 45 cm depth at the sulfatemethane transition zone may be in part due to tracer back flux associated with AOM (see Holler et al., PNAS 2011). Authors Reply: Thank you for this valid point. That peak in hydrogenotrophic methanogenesis is indeed situated at SMTZ, which is why tracer back flux from AOM is possible. We added this information to the discussion part under 4.1.1.

Figure 7: What is the difference between the methane concentration in this figure and in Figure 6? Why not combine Figures 6 & 7?

Authors Reply: Figure 6 and 7 (now 7 and 8) show the results of two different experiments even though both are from September 2014. Figure 7 shows sediment methane concentrations from the 0-1 cmbsf sediment interval over a more detailed sampling period (at least in the first 10 days) after the addition of non-labeled methanol. Figure 6 focuses on a different sediment interval (0-2 cmbsf) and the addition of 13C-labeled methanol with resulting headspace methane content and isotopic composition. We tried to clarify the figure captions.

Lines 525 and following: What are the criteria for calling something a "strong" or "weak" correlation.

Authors Reply: We decided to delete any characterization of "strong' or "weak" correlations in the text, as it is hard to identify correlation strongness with PCA. We therefore focus on positive, negative or zero correlation.

Line 554 It might be good to briefly describe how BES works as an inhibitor, and why it has no effect here.

Authors Reply: Thank you for this helpful comment. We added the function of BES and the possible explanation for BES insensitivity to this paragraph.

Line 566 How deep is bioturbation in Bognis Eck? And was the shell at 20 cm living or just debris?

Authors Reply: From previous studies we know that the bioturbation depth in Eckern-foerde Bay sediments is around 10 cm (e.g. (D'Andrea et al., 1996; Orsi et al., 1996; Bertics et al., 2013; Dale et al., 2013We added the bioturbation depth in line 588. The mollusk shells were empty, and we added this information to the text. It would not be correct in our opinion to call it shells debris, as we cannot be sure if the mollusk was still alive when we collected the core (and died during core storage).

Figure 8: Based on what criteria was 0-5 cm depth for integrated methanogenesis chosen, whereas, similar data, but from 0-25 cm is shown in Figure4?

Authors Reply: In Figure 8 we provide a closer look at methanogenesis directly at the sediment-water interface (0-5 cm), as this layer is likely to be most impacted by water column parameters. Figure 4 on the other hand provides an overview of the total integrated (0-25 cm) surface methanogenesis activity over the sampling period to investigate variations between months.

Line 614: Again, this looks like a growth curve.

Authors Reply: We agree with the reviewer and added this discussion.

Line 637: These organisms became dominant due to the highly enriched methanol concentrations employed. This does not say anything about their importance under in situ conditions.

Authors Reply: We thank the reviewer for this valid point. While we can make assumptions about the initial presence of methylotrophic methanogens under in-situ conditions, we cannot make assumptions about their abundance. We adapted the interpretation accordingly.

Line 690 and following: Changing sulfate concentration-depth profiles as a response to changing salinity conditions indicates that this is a non-steady-state situation. Ergo, it is not possible to use this as an indication of microbial sulfate reduction.

Authors Reply: We are not sure if we understand the comment of the reviewer correctly. What we are trying to say (and which has been shown in other studies) is that due to the close coupling of sulfate to salinity, a decline in salinity would imply a decline in sulfate and hence a faster exhaustion of sulfate in the sediment leaving less organic matter to sulfate reduction. We clarified this part.

Line 841: How does the fueling of AOM above the SMTZ cause methanogenesis to play an "underestimated" role? I would expect that AOM would minimize the impact of methanogenesis on the water column methane budget.

Authors Reply: Thank you for this comment. Yes, AOM would minimize emissions and this is surely the case. But this close link between methanogenesis and AOM has been overlooked so far. We added a few comments on carbon cycling.

Technical comments: Line 138 "that" instead of "which"

Authors Reply: Done

Line 437 "Content" not "concentration" for POC wt%

Authors Reply: Done

Line 612: Sentence is confusing: "of" rather than "if"?

Authors Reply: Done

Also, the population changes to the new conditions; you do not have any evidence for adaption (and evolutionary concept).

[Figure]

Authors Reply: Formulation changed, deleted "adapted".

Please also note the supplement to this comment:
https://www.biogeosciences-discuss.net/bg-2017-36/bg-2017-36-AC1-supplement.pdf

**Supplement:**

**Microbial methanogenesis in the sulfate-reducing zone in sediments from Eckernförde Bay, SW Baltic Sea**

Johanna Maltby[a,b*], Lea Steinle[c,a], Carolin R. Löscher[d,a], Hermann W. Bange[a], Martin A. Fischer[e], Mark Schmidt[a], Tina Treude[a,*]

[a] *GEOMAR Helmholtz Centre for Ocean Research Kiel, Department of Marine Biogeochemistry, 24148 Kiel, Germany*

[b] *Present Address: Natural Sciences Department, Saint Joseph's College, Standish, Maine 04084, USA*

[c] *Department of Environmental Sciences, University of Basel, 4056 Basel, Switzerland*

[d] *Nordic Center for Earth Evolution, University of Southern Denmark, 5230 Odense, Denmark*

[e] *Institute of Microbiology, Christian-Albrecht-University Kiel, 24118 Kiel, Germany*

[e]*Department of Earth, Planetary, and Space Sciences,  University of California Los Angeles (UCLA), Los Angeles, California 90095-1567, USA*

g *Department of Atmospheric and Oceanic Sciences, University of California Los Angeles (UCLA), Los Angeles, California 90095-1567, USA*

*Correspondence: jmaltby@sjcme.edu, ttreude@g.ucla.edu

**Abstract**

Benthic microbial methanogenesis is a known source of methane in marine systems. In most sediments, the majority of methanogenesis is located below the sulfate-reducing zone, as sulfate reducers outcompete methanogens for the major substrates hydrogen and acetate. Coexistence of methanogenesis and sulfate reduction has been shown before and is possible by usage of non-competitive substrates by the methanogens such as methanol or methylated amines. However, the knowledge about magnitude, seasonality and environmental controls on this non-competitive methane production is sparse. In the present study, the The presence of surface methanogenesis (0-30 centimeters below seafloor, cmbsf), located here defined as methanogenesis within the within the sulfate-reducing zonesulfate-rich 
[revised manuscript text omitted]

25-30 cmbsf). It is possible that additional carbon sources led to increased local fermentation processes, for instance from the deposition of macro algae detritus, which is produced during winter storms and can be transported into deeper sediment layers by bioturbation, where it is digested and
released as fecal pellets (Meyer-Reil, 1983; Bertics et al., 2013). Such additional carbon sources from
fresh material could lead to the local accumulation of excess hydrogen through fermentation and
reduce the competition for $H_2$ between sulfate reducers and methanogens (Treude et al., 2009). C/N
ratios in March 2013 were more scattered compared to other months in 2013 and 2014, indicating
the transport of labile material into the sediment. Eckernförde Bay sediments are known for
bioturbation especially during early spring by mollusks and polycheates in the upper 10 cm of the
sediment (D'Andrea et al., 1996; Orsi et al., 1996; Bertics et al., 2013; Dale et al., 2013), and empty
mollusk shells were observed even at depth of ~ 20 cmbsf during sampling in the present study
(personal observation).

Hydrogenotrophic methanogenesis was also detected in the gravity core in September 2013.
Maximum  rates were found at 45 cmbsf and 138 cmbsf, indicating a higher usage
of  $H_2$ at depths > 40 cmbsf, where sulfate was depleted and thus the competition between
sulfate reducers and methanogens was relieved. It should be noted, however, that the peak in
 at 45 cmbsf could  also be a result of tracer ($H^{14}CO_3^-$)
back flux associated with AOM (Holler et al., 2011), as this peak is situated directly at the SMTZ (Fig.
4)

**4.1.2 Inhibition of sulfate reducers**

 Supposedly the competition between methanogens and sulfate reducers within the upper 30
cmbsf led to the predominant utilization of non-competitive substrates by methanogenesis, as
indicated by low hydrogenotrophic vs. higher net methanogenesis rates (see discussion above).
After the addition of the sulfate-reducer inhibitor molybdate, competitive substrates ($H_2$ and
acetate (Oremland & Polcin, 1982; King et al., 1983) were available for methanogenesis ~~as indicated
by(up to 30 times)56~~).
Notably, highest rates in the molybdate treatment were measured at the shallowest sediment depth
at most sampling months (except November 2013), pointing towards the strongest competition
between sulfate reducers and methanogens directly at the top 0-1 cmbsf. Accordingly, maximum
 sulfate reduction activity was detected in this
depth layer in earlier studies (Bertics et al. 2013; Treude et al. 2005). In conclusion, findings from the
molybdate addition experiment highlight that the methanogenic community is subject to a strong
competition with sulfate reducers in the surface sediments and that the majority of the observed
methane production under sulfate-reducing conditions can be attributed to the utilization of non-
competitive substrates.

**4.1.3 Inhibition of methanogenesis by BES**

BES acts as a specific inhibitor of methanogens, because it is a structural analaogue of 2-mercaptoethanesulfonate (coenzyme M), an enzyme only found in methanogens (Gunsalus et al., 1978; Hoehler et al., 1994). Addition of BES did not result in the expected inhibition of potential methanogenesis; instead rates were in the same range as the control treatment (Fig. 76). Consequently, eEither the inhibition of BES was incomplete, or the methanogens were insensitive to BES (Hoehler et al., 1994; Smith & Mah, 1981; Santoro & Konisky, 1987). However, tThe BES concentration used applied in the present study (60 mM) has been shown to result in successful inhibition of methanogens in previous studies (Hoehler et al., 1994). Therefore, the presence of methanogens that are insensitive to BES was is more likely. The insensitivity to BES in methanogens was previously is explained ofby heritable changes in BES permeability or formation of BES-resistant enzymes (Smith & Mah, 1981; Santoro & Konisky, 1987). Such BES resistance was found in *Methanosarcina* mutants (Smith & Mah, 1981; Santoro & Konisky, 1987). This genus was successfully detected in our samples (for more details see 4.1.5), and is known for mediating the methylotrophic pathway (Keltjens & Vogels, 1993), supporting our hypothesis on the utilization of non-competitive substrates by methanogens. Insensitivity to BES in the presented sediments would support the hypothesis that methanogenesis in the sulfate reduction zone is mainly driven via the methylotrophic pathway, as BES resistance was shown in *Methanosarcina* mutants in earlier studies (Smith & Mah, 1981; Santoro & Konisky, 1987). This genus was, a genus which we successfully detected in our samples (for more details see Sect. 4.1.5), and which is known for mediating the methylotrophic pathway (Keltjens & Vogels, 1993).

**4.1.4 Methanol addition**

High potential methanogenesis rates observed after the addition of the non-competitive substrate methanol (Fig. 65) leads to the assumption that methylotrophic methanogens non-competitive substrates relieve the competition between methanogens and sulfate reducersare present in surface sediments of Eckernförde Bay. Except for November 2013, highest rates in the methanol-treatment were detected in the upper 0-5 cmbsf and decreased with depth(Fig. 5). Highest methanogenesis rates in the upper 0-5 cmbsf of the methanol-treatment This observation can be interpreted as followstwofold: (1) The amount availability of non-competitive substrates, including methanol, was most likely highest at the sediment surface, as those substrates are derived from fresh organic matter, such as pectin or betaine and dimethylpropiothetin (both osmoprotectants) (Zinder, 1993). Hence, the methanol-utilizing methanogenic community had it highest abundance in this zone. (2) Sulfate reduction is most dominant in the 0-5 cmbsf (Treude et al., 2005a; Bertics et al., 2013), which probably leads prevalent methanogens to an increased be more adapted to the usage of if non-
competitive substrates.
It should be noted that even though methanogenesis rates were calculated assuming a linear
increase in methane concentration concentration content over the entire incubation to make a
better comparison between different treatments, the methanol treatments generally showed a
delayed response in methane development (Fig. 78, Supplement, Fig. S21). We suggest that this
delayed response was a reflection of cell growth by methanogens utilizing the surplus methanol. We
are therefore unable to decipher whether methanol plays a major role as a substrate in the
Eckernförde Bay sediments compared to possible alternatives, as its concentration is relatively low in
the natural setting (1.05 µM in the 0-1 cmbsf layer, ~1.2 µM at 1-25 cmbsf, June 2014 sampling, G.-C.
Zhuang unpubl. data). It is conceivable that other non-competitive substrates, A similar delay in
methane production was observed in organic-rich surface sediments sampled off Peru and was
explained by the predominant use of alternative non-competitive substrates such as methylated
sulfides (e.g., dimethyl sulfide or methanethiol), are more relevant for the support of surface
methanogenesis (Maltby et al., 2016)). , resulting in a change of methanogenic community after
addition of methanol similar to a growth curve. In the marine environment, dimethyl sulfide mainly
originate from the algae osmoregulatory compound dimethylsulfoniopropionate (DMSP) (Van Der
Maarel & Hansen, 1997), which could have accumulated in Eckernförde Bay sediments, due to
intense sedimentation of algae blooms (Bange et al., 2011). (Maltby et al., 2016) detected a similar
delay in methane production in organic-rich surface sediments sampled off Peru after the addition of
methanol, and suggested the predominant use of methylated sulfides. Certain *Methanosarcina*
species have been shown to use DMSP as a substrate (Sieburth et al., 1993; Van Der Maarel &
Hansen, 1997), a genus, which has been detected in our samples (see 4.1.5 for more details under
Sect. 4.1.5).
Additionally, there are hints that methylated sulfur compounds may be generated through
nucleophilic attack by sulfide on the methyl groups in the sedimentary organic matter (Mitterer,
2010). As shown in the present study, sulfide was an abundant species in the surface sediment (up to
mM levels) (Fig. 1 and 2).
While we are confident that methanol is present in the examined sediments in concentrations
ranging from 0.03 µM up to 1.05 µM in June 2014 (data not shown), with the highest concentration
right at the sediment-water interface, we cannot be sure about the quantity of other non-
competitive substrates. However, the high organic carbon input as well as the high sulfide
concentrations make it very likely that dimethyl sulfide or methanethiol are present.

**4.1.5 Presence of methylotrophic methanogens**

[revised manuscript text omitted]

Sea water and less saline Baltic Sea water (Bange et al., 2011). The PCA detected a  positive correlation between integrated surface methanogenesis (0-5 cmbsf) and salinity in the bottom-near water (Fig. 10a). This correlation can hardly be explained by salinity alone, as methanogens feature a broad salinity range from freshwater to hypersaline (Zinder, 1993). More likely, methanogenesis was affected by variations in water-column sulfate concentrations, which change alongside salinity  (Pattnaik et al., 2000), providing either more (high salinity) or less  (low salinity) of the electron acceptor for the degradation of organic matter by the sulfate-reducing bacteria in the sediment

[revised manuscript text omitted]

was detected  0-25 cmbsf, which was above  the expected steepest increase in methane concentration. Hence, a part of the AOM zone could have been missed during sampling. But the authors concluded that  the activity found was entirely fueled by deep methanogenesis , as  the integrated AOM rates (0.8-1.5 mmol m$^{-2}$

d⁻¹) were in the same  range as the predicted deep methane flux (0.66-1.88 mmol m⁻² d⁻¹) from below the SMTZ .

Together with the data set presented here we postulate that  AOM above the SMTZ (0.8 mmol m⁻² d⁻¹, Treude et al., (2005a) could be  partially or entirely fueled by surface methanogenesis. If, in the extreme scenario, surface methanogenesis would represent the only methane source for  AOM above the SMTZ, then surface methanogenesis is more likely in the range of 0.9 mmol m⁻² d⁻¹ (AOM + net surface methanogenesis). Even though the contribution of surface methanogenesis to surface AOM remains speculative, it leads to the assumption that  surface methanogenesis could play a much bigger role for benthic carbon cycling in the Eckernförde Bay than previously thought. Whether surface methanogenesis at Eckernförde Bay has the potential for the direct emission of methane  into the water column goes beyond the scope of this study and should be tested in the future . In fact, surface methanogenesis was found to correlate with methane concentrations in the water column near the seafloor, but at the same time this could be related to gas ebullition from below the SMTZ, which is likely a more potent methane source to the water column (Fig. 1).

[revised manuscript text omitted]

Gunsalus, R.P., Romesser, J.A. & Wolfe, R.S. (1978). Preparation of coenzyme M analogs and their
activity in the methyl coenzyme M reductase system of Methanobacterium
thermoautotrophicum. *Biochemistry*. 17 (12). pp. 2374–2377.

Hansen, H.-P., Giesenhagen, H.C. & Behrends, G. (1999). Seasonal and long-term control of bottom-
water oxygen deficiency in a stratified shallow-water coastal system. *ICES Journal of Marine*
*Science*. 56. pp. 65–71.

Hartmann, D.L., Klein Tank, A.M.G., Rusticucci, M., Alexander, L.V., Brönnimann, S., Charabi, Y.,
Dentener, F.J., Dlugokencky, D.R., Easterling, D.R., Kaplan, A., Soden, B.J., Thorne, P.W., Wild,
M. & Zhai, P.M. (2013). Observations: Atmosphere and Surface. In: *Climate Change 2013: The*
*pHysical Science Basis. Contribution Group I to the Fifth Assessment Report of the*
*Intergovernmental Panel on Climate Change*. United Kingdom and New York, NY, USA:
Cambridge University Press.

Heyer, J., Hübner, H. & Maaβ, I. (1976). Isotopenfraktionierung des Kohlenstoffs bei der mikrobiellen
Methanbildung. *Isotopenpraxis Isotopes in Environmental and Health Studies*. 12 (5). pp. 202–
205.

Hoehler, T.M., Alperin, M.J., Albert, D.B. & Martens, C.S. (1994). Field and laboratory studies of
methane oxidation in an anoxic marine sediment: Evidence for a methanogen-sulfate reducer
consortium. *Global Biogeochemical Cycles*. 8 (4). pp. 451–463.

Holler, T., Wegener, G., Niemann, H., Deusner, C., Ferdelman, T.G., Boetius, A., Brunner, B. & Widdel,
F. (2011). Carbon and sulfur back flux during anaerobic microbial oxidation of methane and
coupled sulfate reduction. *Proceedings of the National Academy of Sciences of the United States*
*of America*. 108 (52). pp. E1484-90.

[revised manuscript text omitted]

Figures

**Figure 1**

[Figure]

**Figure 1**

[Figure]

**Figure 3<s>2</s>**

[Figure]

**Figure
[Figure]
**

[Figure]

 **Figure 54**

[Figure]

**Figure 6**5

[Figure]

**Figure 76**

[Figure]

[Figure]

**Figure 87**

[Figure]

**Figure 98**

[Figure]

**Figure 109**

[Figure]

**Figure 1_10**

a

[Figure]

b

[Figure]

---

## Author Comment (AC2) · 14 Sep 2017

We would like to thank the reviewer for her/his critical comments, which we think helped to improve the quality and clarity of this manuscript. We hope our responses and adaptations are adequate to accept this manuscript for publication in Biogeosciences. Please find our detailed responses below. Anonymous Referee #2

The work presented by Maltby et al. is really nice piece of study gathering results from several impressive campaign of sampling and involving different cutting-edge methods.

[Figure]

Their findings give an interesting overview of biological processes and environmental factors controlling methane emissions from sediments and water column of a Baltic sea bay, well known for its importance in global methane emissions. The originality of their work lies in the demonstration of co-occurence of sulphate reduction and methanogenesis in surface sediments. This co-existence is permitted by a mechanism developed by some methanogenesis microorganisms to escape from the strong competition with sulphate-reducing microorganisms: using (releasing? I did not find information on that) non-competitive substrates. The manuscript is overall well written except the abstract, see my comments below. I have only two main concerns.

First, the article is sometime written in a way that only initiates of the field may touch. The first sentence of abstract directly starts with the work done without putting the study in a wider context. The object you study is complex and well structured. We do not immediately understand the relevance of studying methanogenesis in the sulphatereducing zone.

Authors Reply: We thank the reviewer for this comment. We agree that the abstract starts abrupt and therefore included a short introduction to explain the reasoning for this study.

We neither understand that you studied surface, deep sediment and the water column and not only surface (sulphate-reducing) sediment. The reference of "a noncompetitive substrate" is not understandable. Which competition do you refer ? Implying which organisms?

Authors Reply: We hope that these questions are answered in the abstract now after we added some introduction and definitions.

In the introduction, it could be useful to build a synthetic figure summarizing the studied ecosystem including the different compartments, different organisms, interactions among organisms, exchanges of matters between these compartments.

Authors Reply: We appreciate this idea and added a scheme (see new Fig. 1).

Second, the (minor) contribution of surface methanogenesis to total methane emissions from this ecosystem is a bit hidden in the article. This contribution deserves to be clearly presented in the abstract. To my point of view, the minor contribution of this mechanism does not question the quality and relevance of this study, and is an important information.

Authors Reply: We understand that the role of surface methanogenesis to total benthic methane emissions is not discussed in length in this paper. However, to be able to go into more detail about the contribution of surface methanogenesis, more detailed studies about emission rates into the water column, methane oxidation rates in the sediment (oxic and anoxic) as well as in the water column would be necessary. Our presented data set gives a first idea about surface methanogenesis and its potential role, but anything beyond that would be speculative.

In the same vein, the statement that surface methanogenesis could play a key role in fueling the surface anaerobic oxidation of methane is speculative since this last process was not measured in the study.

Authors Reply: We agree that the statement is speculative, as AOM was not determined in the present study. But we have good indications of surface AOM from radiotracer incubations from previous studies (Treude et al. 2005) in the Eckernfoerde Bay. Furthermore, we see no issue with formulating a hypothesis.

Specific comments. Line 30 supress "in the manipulated experiments.

Authors Reply: Done.

L31-33 this new objective that pops up too late. Please gather your objective in one sentence

Authors Reply: We gathered the objectives.

L47 replace "makes an important contribute" by "substantially contributes to". I did not understand the last part "as it could..."

Authors Reply: We changed the sentence to make it clearer.

L78-79 and throughout the manuscript. The expression "Environmental control mechanisms" is a bit elusive. Do you mean "environmental controls" or "biological processes"? Try to better specify what should be better studied.

Authors Reply: We changed it to environmental controls wherever we used "environmental control mechanisms".

L164 Rewrite your sentence to clarify. Could be "Biological activities of samples were stopped by the addition of mercury chloride solution..."

Authors Reply: Done.

L177 "extracted"? you mean "sampled" for analysis?

Authors Reply: "porewater extraction" is a commonly used expression in sediment geochemistry.

L193-216 I am not expert in measurement of methane concentration in sediment, but I am wondering whether the fact of cutting sediment core in 1 cm sediment interval could release, at least a part of, the methane you wish to quantify.

Authors Reply: It is correct that some methane in form of gas bubbles might escape during the slicing procedure as it does during multicoring. However, we accept that fact of potential methane loss, as these gas bubbles originate from deep methanogenesis below the sulfate-reducing zone. We are more interested in the methane kept in the tiny spaces between the shallow sediment, which we indeed can determine with the presented method.

L236 Could you rapidly explain again what is the hydrogenotrophic methanogenesis? And what is the interest of measuring this in the context of your study?

Authors Reply: Done.

L389 Supress "in" before september.

Authors Reply: Done.

L401 I guess you're talking about the C/N of particulate organic matter, but I am not sure. Please specify.

Authors Reply: Done.

L543 I propose you to replace the end of your sentence by "...the following observations that will be discussed in more detail in the following chapters". My first reaction was to try to understand your arguments before reading the following chapters.

Authors Reply: Done. Thanks for pointing that out.

L569 Your explanations about the competition between sulphate-reducing and methanogenesis microorganisms, and the strategy of methylotrophic methatogeners to escape from this competition, are very clear and convincing. Now I am wondering whether there is competition between hydrogenotrophic and methyltrophic microorganisms. And if yes, does this competition change with depth?

Authors Reply: To our knowledge, there is no competition between hydrogenotrophic and methylotrophic methanogenesis, as these metabolism are using different substrates ( that is why this categorization makes sense). Some hydrogenotrophic methanogens are able to use secondary alcohols in addition, however, they normally cannot use the substrates of methylotrophic methanogens (methanol, methylated amines, dimethylsulfides).

L587-592 This sentence is too long. Split your explanations into 2 sentences.

Authors Reply: Done.

L614 If I follow well, you should add a "P" after "DMS".

Authors Reply: Done.

L605-631 Maybe this is a limit of your study of not having quantified some key non-competitive substrates in sediments and water. It could be discussed in a paragraph drawing next investigations that could be done.

Authors Reply: We agree with the reviewer that this is a limit of our study. Our focus in the present study was to see the potential rates when fueled with non-competitive substrates such as methanol. The next step would be to measure the natural concentrations. We added a paragraph at the end of 4.1.4 to advise the reader that we actually did not measure most of the non-competitive substrates (besides methanol in June 2014).

L640 What fractionation are you discussing? An isotopic fractionation? You must better explain.

Authors Reply: Changed to isotopic fractionation.

L488-489 Could you check whether such moderate isotopic fractionation (factor of 1.07-1.08) could explain an increase of delta of almost 200 per mille. I have a doubt.

Authors Reply: We used fractionation factors from previous studies using non-13C-spiked methanol. With our data set we were not able to calculate a reliable fractionation factor, as you have to use concentrations of substrate and product from the stationary phase. However, the methanol in our experiments was used up before a stationary phase was observed (see Fig. 6 (now 7). At timepoint 2 (14 days) the fractionation factor was 1.22 (calculated by authors), but this value should not be used, as it was retrieved from the slope phase. To actually be able to determine the fractionation factor in our case, more experiments would have to be conducted.

L644-645 This sentence is not clear. What would be the alternative explanation?

Authors Reply: Revised the sentence, as there is no other explanation.

[Figure]

L646 One bracket is lacking at the end of sentence.

Authors Reply: Done.

L684-694 I did not understand your explanations. Please try to reformulate and be more direct when you propose an interpretation.

Authors Reply: Revised the sentences.

L706-707 Did you find results going in this way as well?

Authors Reply: As described in this paragraph, the PCA did indeed show a positive correlation between methanogenesis and organic carbon availability, supporting the idea that organic matter availability can lead to coexistence of sulfate reduction and methanogenesis. To really answer your question, we would also need results from less-organic-rich systems, such as open ocean sediments.

L713-736 This section is really too long. Split it in two paragraphs, one focusing on the effect of POC amount and the other on C/N ratio.

Authors Reply: As POC and C/N are directly interlinked, we would prefer to discuss both factors together. We think it is necessary to discuss the availability of POC and the freshness of POC together. However, we formulated subtitles to make this paragraph easier to read.

L830-831 I do not understand your interpretation of the positive correlation between surface methanogenesis and C/N ratio of POM.

Authors Reply: We changed the sentence a little bit to make it more clear. Essentially, a positive correlation means higher methanogenesis when C/N ratio is higher, thus indicating less-fresh material. Usually, with increasing sediment depth the organic material becomes less-fresh (= higher C/N ratio). But at these depths, fermentation processes take place, which then provide substrates to the methanogens. So it is not surprising that methanogenesis is high at deeper depth.

L805-809 and 832-834. This process of anaerobic consumption of methane (AOM) was not measured in this study making all these discussions around the key role of surface methanogenesis in fuelling AOM very speculative. I do not understand why deep methanogenesis, which contributes for the major part of methane emissions, does not contribute to AOM fueling. It sounds like you would absolutely like to give acentral importance to surface methanogenesis.

Authors Reply: We agree that this is a speculative statement. But we are confident making this connection between surface methanogenesis and surface AOM, as it was measured in similar sediments from the same sampling area. However, we softened our conclusion and provided more details for our reasoning.

Please also note the supplement to this comment:
https://www.biogeosciences-discuss.net/bg-2017-36/bg-2017-36-AC2-supplement.pdf

**Supplement:**

**Microbial methanogenesis in the sulfate-reducing zone in sediments from Eckernförde Bay, SW Baltic Sea**

Johanna Maltby[a,b*], Lea Steinle[c,a], Carolin R. Löscher[d,a], Hermann W. Bange[a], Martin A. Fischer[e], Mark Schmidt[a], Tina Treude[a,*]

[a] *GEOMAR Helmholtz Centre for Ocean Research Kiel, Department of Marine Biogeochemistry, 24148 Kiel, Germany*

[b] *Present Address: Natural Sciences Department, Saint Joseph's College, Standish, Maine 04084, USA*

[c] *Department of Environmental Sciences, University of Basel, 4056 Basel, Switzerland*

[d] *Nordic Center for Earth Evolution, University of Southern Denmark, 5230 Odense, Denmark*

[e] *Institute of Microbiology, Christian-Albrecht-University Kiel, 24118 Kiel, Germany*

[e]*Department of Earth, Planetary, and Space Sciences,  University of California Los Angeles (UCLA), Los Angeles, California 90095-1567, USA*

g *Department of Atmospheric and Oceanic Sciences, University of California Los Angeles (UCLA), Los Angeles, California 90095-1567, USA*

*Correspondence: jmaltby@sjcme.edu, ttreude@g.ucla.edu

**Abstract**

Benthic microbial methanogenesis is a known source of methane in marine systems. In most sediments, the majority of methanogenesis is located below the sulfate-reducing zone, as sulfate reducers outcompete methanogens for the major substrates hydrogen and acetate. Coexistence of methanogenesis and sulfate reduction has been shown before and is possible by usage of non-competitive substrates by the methanogens such as methanol or methylated amines. However, the knowledge about magnitude, seasonality and environmental controls on this non-competitive methane production is sparse. In the present study, the The presence of surface methanogenesis (0-30 centimeters below seafloor, cmbsf), located here defined as methanogenesis within the within the sulfate-reducing zonesulfate-rich 
[revised manuscript text omitted]

25-30 cmbsf). It is possible that additional carbon sources led to increased local fermentation processes, for instance from the deposition of macro algae detritus, which is produced during winter storms and can be transported into deeper sediment layers by bioturbation, where it is digested and
released as fecal pellets (Meyer-Reil, 1983; Bertics et al., 2013). Such additional carbon sources from
fresh material could lead to the local accumulation of excess hydrogen through fermentation and
reduce the competition for $H_2$ between sulfate reducers and methanogens (Treude et al., 2009). C/N
ratios in March 2013 were more scattered compared to other months in 2013 and 2014, indicating
the transport of labile material into the sediment. Eckernförde Bay sediments are known for
bioturbation especially during early spring by mollusks and polycheates in the upper 10 cm of the
sediment (D'Andrea et al., 1996; Orsi et al., 1996; Bertics et al., 2013; Dale et al., 2013), and empty
mollusk shells were observed even at depth of ~ 20 cmbsf during sampling in the present study
(personal observation).

Hydrogenotrophic methanogenesis was also detected in the gravity core in September 2013.
Maximum  rates were found at 45 cmbsf and 138 cmbsf, indicating a higher usage
of  $H_2$ at depths > 40 cmbsf, where sulfate was depleted and thus the competition between
sulfate reducers and methanogens was relieved. It should be noted, however, that the peak in
 at 45 cmbsf could  also be a result of tracer ($H^{14}CO_3^-$)
back flux associated with AOM (Holler et al., 2011), as this peak is situated directly at the SMTZ (Fig.
4)

**4.1.2 Inhibition of sulfate reducers**

 Supposedly the competition between methanogens and sulfate reducers within the upper 30
cmbsf led to the predominant utilization of non-competitive substrates by methanogenesis, as
indicated by low hydrogenotrophic vs. higher net methanogenesis rates (see discussion above).
After the addition of the sulfate-reducer inhibitor molybdate, competitive substrates ($H_2$ and
acetate (Oremland & Polcin, 1982; King et al., 1983) were available for methanogenesis ~~as indicated
by(up to 30 times)56~~).
Notably, highest rates in the molybdate treatment were measured at the shallowest sediment depth
at most sampling months (except November 2013), pointing towards the strongest competition
between sulfate reducers and methanogens directly at the top 0-1 cmbsf. Accordingly, maximum
 sulfate reduction activity was detected in this
depth layer in earlier studies (Bertics et al. 2013; Treude et al. 2005). In conclusion, findings from the
molybdate addition experiment highlight that the methanogenic community is subject to a strong
competition with sulfate reducers in the surface sediments and that the majority of the observed
methane production under sulfate-reducing conditions can be attributed to the utilization of non-
competitive substrates.

**4.1.3 Inhibition of methanogenesis by BES**

BES acts as a specific inhibitor of methanogens, because it is a structural analaogue of 2-mercaptoethanesulfonate (coenzyme M), an enzyme only found in methanogens (Gunsalus et al., 1978; Hoehler et al., 1994). Addition of BES did not result in the expected inhibition of potential methanogenesis; instead rates were in the same range as the control treatment (Fig. 76). Consequently, eEither the inhibition of BES was incomplete, or the methanogens were insensitive to BES (Hoehler et al., 1994; Smith & Mah, 1981; Santoro & Konisky, 1987). However, tThe BES concentration used applied in the present study (60 mM) has been shown to result in successful inhibition of methanogens in previous studies (Hoehler et al., 1994). Therefore, the presence of methanogens that are insensitive to BES was is more likely. The insensitivity to BES in methanogens was previously is explained ofby heritable changes in BES permeability or formation of BES-resistant enzymes (Smith & Mah, 1981; Santoro & Konisky, 1987). Such BES resistance was found in *Methanosarcina* mutants (Smith & Mah, 1981; Santoro & Konisky, 1987). This genus was successfully detected in our samples (for more details see 4.1.5), and is known for mediating the methylotrophic pathway (Keltjens & Vogels, 1993), supporting our hypothesis on the utilization of non-competitive substrates by methanogens. Insensitivity to BES in the presented sediments would support the hypothesis that methanogenesis in the sulfate reduction zone is mainly driven via the methylotrophic pathway, as BES resistance was shown in *Methanosarcina* mutants in earlier studies (Smith & Mah, 1981; Santoro & Konisky, 1987). This genus was, a genus which we successfully detected in our samples (for more details see Sect. 4.1.5), and which is known for mediating the methylotrophic pathway (Keltjens & Vogels, 1993).

**4.1.4 Methanol addition**

High potential methanogenesis rates observed after the addition of the non-competitive substrate methanol (Fig. 65) leads to the assumption that methylotrophic methanogens non-competitive substrates relieve the competition between methanogens and sulfate reducersare present in surface sediments of Eckernförde Bay. Except for November 2013, highest rates in the methanol-treatment were detected in the upper 0-5 cmbsf and decreased with depth(Fig. 5). Highest methanogenesis rates in the upper 0-5 cmbsf of the methanol-treatment This observation can be interpreted as followstwofold: (1) The amount availability of non-competitive substrates, including methanol, was most likely highest at the sediment surface, as those substrates are derived from fresh organic matter, such as pectin or betaine and dimethylpropiothetin (both osmoprotectants) (Zinder, 1993). Hence, the methanol-utilizing methanogenic community had it highest abundance in this zone. (2) Sulfate reduction is most dominant in the 0-5 cmbsf (Treude et al., 2005a; Bertics et al., 2013), which probably leads prevalent methanogens to an increased be more adapted to the usage of if non-
competitive substrates.
It should be noted that even though methanogenesis rates were calculated assuming a linear
increase in methane concentration concentration content over the entire incubation to make a
better comparison between different treatments, the methanol treatments generally showed a
delayed response in methane development (Fig. 78, Supplement, Fig. S21). We suggest that this
delayed response was a reflection of cell growth by methanogens utilizing the surplus methanol. We
are therefore unable to decipher whether methanol plays a major role as a substrate in the
Eckernförde Bay sediments compared to possible alternatives, as its concentration is relatively low in
the natural setting (1.05 µM in the 0-1 cmbsf layer, ~1.2 µM at 1-25 cmbsf, June 2014 sampling, G.-C.
Zhuang unpubl. data). It is conceivable that other non-competitive substrates, A similar delay in
methane production was observed in organic-rich surface sediments sampled off Peru and was
explained by the predominant use of alternative non-competitive substrates such as methylated
sulfides (e.g., dimethyl sulfide or methanethiol), are more relevant for the support of surface
methanogenesis (Maltby et al., 2016)). , resulting in a change of methanogenic community after
addition of methanol similar to a growth curve. In the marine environment, dimethyl sulfide mainly
originate from the algae osmoregulatory compound dimethylsulfoniopropionate (DMSP) (Van Der
Maarel & Hansen, 1997), which could have accumulated in Eckernförde Bay sediments, due to
intense sedimentation of algae blooms (Bange et al., 2011). (Maltby et al., 2016) detected a similar
delay in methane production in organic-rich surface sediments sampled off Peru after the addition of
methanol, and suggested the predominant use of methylated sulfides. Certain *Methanosarcina*
species have been shown to use DMSP as a substrate (Sieburth et al., 1993; Van Der Maarel &
Hansen, 1997), a genus, which has been detected in our samples (see 4.1.5 for more details under
Sect. 4.1.5).
Additionally, there are hints that methylated sulfur compounds may be generated through
nucleophilic attack by sulfide on the methyl groups in the sedimentary organic matter (Mitterer,
2010). As shown in the present study, sulfide was an abundant species in the surface sediment (up to
mM levels) (Fig. 1 and 2).
While we are confident that methanol is present in the examined sediments in concentrations
ranging from 0.03 µM up to 1.05 µM in June 2014 (data not shown), with the highest concentration
right at the sediment-water interface, we cannot be sure about the quantity of other non-
competitive substrates. However, the high organic carbon input as well as the high sulfide
concentrations make it very likely that dimethyl sulfide or methanethiol are present.

**4.1.5 Presence of methylotrophic methanogens**

[revised manuscript text omitted]

Sea water and less saline Baltic Sea water (Bange et al., 2011). The PCA detected a  positive correlation between integrated surface methanogenesis (0-5 cmbsf) and salinity in the bottom-near water (Fig. 10a). This correlation can hardly be explained by salinity alone, as methanogens feature a broad salinity range from freshwater to hypersaline (Zinder, 1993). More likely, methanogenesis was affected by variations in water-column sulfate concentrations, which change alongside salinity  (Pattnaik et al., 2000), providing either more (high salinity) or less  (low salinity) of the electron acceptor for the degradation of organic matter by the sulfate-reducing bacteria in the sediment

[revised manuscript text omitted]

was detected  0-25 cmbsf, which was above  the expected steepest increase in methane concentration. Hence, a part of the AOM zone could have been missed during sampling. But the authors concluded that  the activity found was entirely fueled by deep methanogenesis , as  the integrated AOM rates (0.8-1.5 mmol m$^{-2}$

d⁻¹) were in the same  range as the predicted deep methane flux (0.66-1.88 mmol m⁻² d⁻¹) from below the SMTZ .

Together with the data set presented here we postulate that  AOM above the SMTZ (0.8 mmol m⁻² d⁻¹, Treude et al., (2005a) could be  partially or entirely fueled by surface methanogenesis. If, in the extreme scenario, surface methanogenesis would represent the only methane source for  AOM above the SMTZ, then surface methanogenesis is more likely in the range of 0.9 mmol m⁻² d⁻¹ (AOM + net surface methanogenesis). Even though the contribution of surface methanogenesis to surface AOM remains speculative, it leads to the assumption that  surface methanogenesis could play a much bigger role for benthic carbon cycling in the Eckernförde Bay than previously thought. Whether surface methanogenesis at Eckernförde Bay has the potential for the direct emission of methane  into the water column goes beyond the scope of this study and should be tested in the future . In fact, surface methanogenesis was found to correlate with methane concentrations in the water column near the seafloor, but at the same time this could be related to gas ebullition from below the SMTZ, which is likely a more potent methane source to the water column (Fig. 1).

[revised manuscript text omitted]

Gunsalus, R.P., Romesser, J.A. & Wolfe, R.S. (1978). Preparation of coenzyme M analogs and their
activity in the methyl coenzyme M reductase system of Methanobacterium
thermoautotrophicum. *Biochemistry*. 17 (12). pp. 2374–2377.

Hansen, H.-P., Giesenhagen, H.C. & Behrends, G. (1999). Seasonal and long-term control of bottom-
water oxygen deficiency in a stratified shallow-water coastal system. *ICES Journal of Marine*
*Science*. 56. pp. 65–71.

Hartmann, D.L., Klein Tank, A.M.G., Rusticucci, M., Alexander, L.V., Brönnimann, S., Charabi, Y.,
Dentener, F.J., Dlugokencky, D.R., Easterling, D.R., Kaplan, A., Soden, B.J., Thorne, P.W., Wild,
M. & Zhai, P.M. (2013). Observations: Atmosphere and Surface. In: *Climate Change 2013: The*
*pHysical Science Basis. Contribution Group I to the Fifth Assessment Report of the*
*Intergovernmental Panel on Climate Change*. United Kingdom and New York, NY, USA:
Cambridge University Press.

Heyer, J., Hübner, H. & Maaβ, I. (1976). Isotopenfraktionierung des Kohlenstoffs bei der mikrobiellen
Methanbildung. *Isotopenpraxis Isotopes in Environmental and Health Studies*. 12 (5). pp. 202–
205.

Hoehler, T.M., Alperin, M.J., Albert, D.B. & Martens, C.S. (1994). Field and laboratory studies of
methane oxidation in an anoxic marine sediment: Evidence for a methanogen-sulfate reducer
consortium. *Global Biogeochemical Cycles*. 8 (4). pp. 451–463.

Holler, T., Wegener, G., Niemann, H., Deusner, C., Ferdelman, T.G., Boetius, A., Brunner, B. & Widdel,
F. (2011). Carbon and sulfur back flux during anaerobic microbial oxidation of methane and
coupled sulfate reduction. *Proceedings of the National Academy of Sciences of the United States*
*of America*. 108 (52). pp. E1484-90.

[revised manuscript text omitted]

Figures

**Figure 1**

[Figure]

**Figure 1**

[Figure]

**Figure 3<s>2</s>**

[Figure]

**Figure
[Figure]
**

[Figure]

 **Figure 54**

[Figure]

**Figure 6**5

[Figure]

**Figure 76**

[Figure]

[Figure]

**Figure 87**

[Figure]

**Figure 98**

[Figure]

**Figure 109**

[Figure]

**Figure 1_10**

a

[Figure]

b

[Figure]

---

## Author Response (AR1)

We would like to thank the editor for the critical comments, which we think helped to improve the quality and clarity of this manuscript. We hope our responses and adaptations are adequate to accept this manuscript for publication in Biogeosciences. Please find our detailed responses below.

**Associate Editor Comment (21. September 2017)**

Dear Authors,

I went through your new version of manuscript and appreciate the modifications you made in order to clarify presentation of your ideas, moderate some statements and delimit limits of your study. However, I would ask you to make an effort to synthetize your discussions by shortening the sections that are viewed as speculative by the two referees. This important effort will avoid a dilution of the main results of your study in speculative discussions using the conditional form.

Regards,

Sébastien Fontaine

**Authors Reply: We agree with the editor/reviewers that the discussion was, in some parts, too repetitive and speculative. In order to follow your suggestions, following adaptations were made:**

1. **We drastically shortened the first part of the discussion (4.1.) to reduce the amount of repetition from the results and the amount of unwarranted speculation.**
2. **In the discussion part 4.3, we deleted the paragraph about our calculation of deep methanogenesis, which was criticized of being too speculative. As we did not actively measure deep methanogenesis in the present study, we agree that this part might go above the scope of our research focus. We further deleted comparisons with other environments (including Table 2) in this chapter, to stay focused on the study site.**
3. **In part 4.3, we were able to add proof to our hypothesis of close coupling between AOM and methanogenesis in surface sediments, which decreases the degree of speculation of this statement. This cryptic methane cycling has been recently demonstrated in labeling incubations with surface sediments from the close-by Aarhus Bay, Denmark (Xiao et al, 2017), and thus could very likely occur in sediments from Eckernfoerde Bay.**
4. **We shortened the summary part to decrease unnecessary repetition.**
5. **We further decided to provide a more concise definition of "surface methanogenesis" by introducing the term "SRZ methanogenesis" (i.e. methanogenesis in the sulfate reduction zone) opposite to ("deep") methanogenesis below the sulfate methane transition zone (SMTZ). We think these terms are more scientifically correct and avoid confusion with the term "sediment surface" in general.**

[revised manuscript text omitted]

than $H_2$. One exemption was detected in the March 2013 incubation, where rates of
hydrogenotrophic methanogenesis exceeded net methanogenesis in discrete depths (5-6 cmbsf and
25-30 cmbsf). It is possible that additional carbon sources led to increased local fermentation
processes, for instance from the deposition of macro algae detritus, which is produced during winter
storms and can be transported into deeper sediment layers by bioturbation, where it is digested and
released as fecal pellets (Meyer-Reil, 1983; Bertics et al., 2013). Such additional carbon sources from
fresh material could lead to the local accumulation of excess hydrogen through fermentation and
reduce the competition for $H_2$ between sulfate reducers and methanogens (Treude et al., 2009). C/N
ratios in March 2013 were more scattered compared to other months in 2013 and 2014, indicating
the transport of labile material into the sediment. Eckernförde Bay sediments are known for
bioturbation especially during early spring by mollusks and polycheates in the upper 10 cm of the
sediment (D'Andrea et al., 1996; Orsi et al., 1996; Bertics et al., 2013; Dale et al., 2013), and empty
mollusk shells were observed even at depth of ~ 20 cmbsf during sampling in the present study
(personal observation).
Hydrogenotrophic methanogenesis was also detected in the gravity core in September 2013.
Maximum rates were found at 45 cmbsf and 138 cmbsf, indicating a higher usage of $H_2$ at depths >
40 cmbsf, where sulfate was depleted and thus the competition between sulfate reducers and
methanogens was relieved. It should be noted, however, that the peak in at 45 cmbsf could also be a
result of tracer ($H^{14}CO_3^-$) back flux associated with AOM (Holler et al., 2011), as this peak is situated
directly at the SMTZ (Fig. 4)

4.1.2 Inhibition of sulfate reducers
Supposedly the competition between methanogens and sulfate reducers within the upper 30 cmbsf
led to the predominant utilization of non-competitive substrates by methanogenesis, as indicated by
lower hydrogenotrophic vs. higher net methanogenesis rates (see discussion above). After the
addition of the sulfate-reducer inhibitor molybdate, competitive substrates ($H_2$ and acetate (Oremland & Polcin, 1982; King et al., 1983) were available for methanogenesis resulting in the (up to 30 times) increase in potential activity (Fig. 6 and 7). Notably, highest rates in the molybdate treatment were measured at the shallowest sediment depth at most sampling months (except November 2013), pointing towards the strongest competition between sulfate reducers and methanogens directly at the top 0-1 cmbsf. Accordingly, maximum sulfate reduction activity was detected in this depth layer in earlier studies (Bertics et al. 2013; Treude et al. 2005). In conclusion, findings from the molybdate addition experiment highlight that the methanogenic community is subject to a strong competition with sulfate reducers in the surface sediments and that the majority of the observed methane production under sulfate-reducing conditions can be attributed to the utilization of non-competitive substrates.

**4.1.3 Inhibition of methanogenesis by BES**

BES acts as a specific inhibitor of methanogens, because it is a structural analaogue of (coenzyme M), an enzyme only found in methanogens (Gunsalus et al., 1978; Hoehler et al., 1994). 
[revised manuscript text omitted]

Figures

**Figure 1**

[Figure]

 **Figure**

[Figure]

**Figure 3**

[Figure]

**Figure 4**

[Figure]

**Figure 5**

[Figure]

[Figure]

**Figure 7**

[Figure]

[Figure]

**Figure 8**

[Figure]

**Figure 9**

[Figure]

**Figure 10**

[Figure]

**Figure 11**

a

[Figure]

b

[Figure]

---

## Author Response (AR2)

**We would like to thank the editor for the recommendations, which we think helped to improve the quality and clarity of this manuscript. We hope our responses and adaptations are adequate to accept this manuscript for publication in Biogeosciences. Please find our detailed responses below.**

**Associate Editor Comment (8. November 2017)**

Dear authors,

I checked your modifications and I am satisfied by your efforts of synthesis. I have some additional recommendations:

- Your manuscript includes too many items (figures and tables). Please select six items that will be presented in the main text and prepare supplementary files for the other items.

- Your discussion has too many subtitles. Could you simplify the structuration by, for example, including some titles in the paragraph? To make it visible you can format it in bold or italic.

Thank you for having submitted to Biogeosciences this nice piece of work.

Sébastien

**Authors Reply:**
- **Even though we are aware that we have many figures, we do think that all of them are necessary, which is why we would like to keep them in the manuscript instead of moving some to the supplement.**

- **We deleted the subheadings (3-level) in the discussion part, as well as the 4-level subheadings in the Material and Methods Part. Instead, we converted them into titles in bold and italic to make the separation between paragraphs visible.**

- **We added one reference in the paper (introduction part, line 105)**

[revised manuscript text omitted]

a

[Figure]

b

[Figure]